**Hot regions of labile and stable soil organic carbon in Germany - Spatial**
**variability and driving factors**
Cora Vos[1], Angélica Jaconi[1], Anna Jacobs[1], Axel Don[1]
[1]Thünen Institute of Climate-Smart Agriculture, Bundesallee 50, 38116 Braunschweig, Germany
Corresponding Author: Axel Don, axel.don@thuenen.de, Tel. +49 531 596 2641
Keywords: Soil organic carbon fractions, near-infrared spectroscopy, NIRS, soil carbon stability,
National Soil Inventory, German Agricultural Soil Inventory, carbon sequestration

## Abstract

Atmospheric carbon dioxide levels can be mitigated by sequestering carbon in the soil. Sequestration can be facilitated by agricultural management, but its influence is not the same on all soil carbon pools, as labile pools with high turnover may be accumulated much faster, but are also more vulnerable to losses. The aims of this study were to 1) assess how soil organic carbon (SOC) is distributed among SOC fractions on national scale in Germany, 2) identify factors influencing this distribution and 3) identify regions with high vulnerability to SOC losses. The SOC content and proportion of two different SOC fractions were estimated for more than 2500 mineral topsoils (<87 g kg$^{-1}$ SOC) covering Germany, using near-infrared reflectance spectroscopy. Drivers of the spatial variability in SOC fractions were determined using the machine learning algorithm cforest. The SOC content and proportions of fractions were predicted with good accuracy (SOC content: $R^2$=0.87-0.90, SOC proportions $R^2$=0.83, ratio of performance to deviation (RPD) 2.4-3.2). Main explanatory variables for distribution of SOC among the fractions were soil texture, bulk soil C/N ratio, total SOC content and pH. For some regions, the drivers were linked to the land-use history of the sites.

Arable topsoils in central and southern Germany were found to contain the highest proportions and contents of stable SOC fractions, and therefore have the lowest vulnerability to SOC losses. North-Western Germany contains an area of sandy soils with unusually high SOC contents and high proportions of light SOC fractions, which are commonly regarded as representing a labile carbon pool. This is true for the former peat soils in this area, which have already lost and are at high risk of losing high proportions of their SOC stocks. Those "black sands" can, however, also contain high amounts of stable SOC due to former heathland vegetation, and need to be treated and discussed separately from 'normal' agricultural soils. Overall, it was estimated that, in large areas all over Germany, over 30% of is stored in easily mineralisable forms. Thus, SOC-conserving management of arable soils in these regions is of great importance.

## 1 Introduction

There is increasing interest in soil organic carbon (SOC) in agricultural soils, as it contributes to soil fertility and also to mitigation of climate change when organic carbon sequestration is enhanced (Post and Kwon, 2000). In agricultural systems the pathway of atmospheric carbon to SOC is controlled by land-use and agronomic management. However, SOC comprises a large range of compounds, ranging from recently added organic matter, such as root litter and exudates, to highly condensed and transformed organic matter that may even be derived from the geogenic parent material. These different compound classes are stabilised in different ways and therefore have different turnover times (Lehmann and Kleber, 2015). Although SOC is now considered as having a continuum of turnover times, it is mostly described and modelled as consisting of different pools that vary in their turnover time (e.g. labile pool, intermediate pool and stabilised pool). The effects of land-use and management are not the same for all soil organic matter compounds,  they differ between SOC pools. Chimento et al. (2016) for example, found that cultivation of perennial woody bioenergy crops increased SOC stocks compared to other bioenergy crops, but the new SOC accumulated only in the light and presumably labile particulate organic matter (POM) fraction. Poeplau and Don (2013a), on the other hand, found that cropland sites that where changed to grassland also sequestered new SOC, but mainly in the more stable fractions. Therefore, the different SOC pools need to be assessed separately from the bulk SOC when discussing the influence of land-use and management on stabilisation and storage of SOC.

One method for experimental quantification of the distribution of SOC among different SOC pools is fractionation. Various fractionation procedures for quantifying SOC fractions have been developed, mostly aiming at isolating fractions with differing turnover times (Poeplau *et al*., in review, Zimmermann et al., 2007a). Determining the distribution of SOC among fractions with assumedly different  turnover times is one step towards understanding the factors influencing SOC stabilisation. All methods for carbon fractionation are quite laborious, time-consuming and therefore expensive, and not feasible for large datasets. Therefore, few studies exist on SOC fractions at regional scale,

indicating a need for development of more efficient methods to predict carbon fractions in
assessment of large datasets. Near-infrared reflectance spectroscopy (NIRS) and mid-infrared
spectroscopy (MIRS), in combination with chemometric methods, have been applied successfully to
predict carbon fractions (Zimmermann *et al.*, 2007b; Baldock *et al.*, 2013; Cozzolino & Moro, 2006;
Reeves *et al.*, 2006). Thus, since prediction of SOC fractions has been demonstrated to be possible
using spectroscopic methods, it should also be possible to go beyond small datasets at field scale in
order to examine how SOC fractions are distributed regionally and the factors that drive this
distribution.
Some impact factors are consistently reported as being important at site scale for the distribution of
SOC among different fractions or pools, one of which is land-use. For Western European croplands
and grasslands, it was shown that a similarly high share of bulk SOC is attributed to fractions
regarded as stable, while in forest soils, a higher proportion of SOC is attributed to more labile SOC
fractions (John *et al.*, 2005; Helfrich *et al.*, 2006; Wiesmeier *et al.*, 2014). Tillage can also have an
impact on SOC pools, as some studies report higher levels of bulk SOC under no-till conditions
compared with conventional tillage, with the majority of this increase occurring in the more labile
carbon pools (Chan et al., 2002; Devine et al., 2014; Liu et al., 2014). This may, however, be just an
effect of carbon redistribution in the soil and not lead to a net increase of SOC (Baker et al., 2007;
Luo et al., 2010).
Fewer studies have examined the SOC distribution into fractions at regional scale and even fewer
have examined factors affecting the proportions of SOC distributed among different fractions or
pools. Wiesmeier *et al.* (2014) determined the distribution of SOC fractions among 99 Bavarian soils
under different land-uses using the fractionation scheme devised by Zimmermann *et al.* (2007a),
which is a combination of particle size and density fractionation. They found that approximately 90%
of the bulk SOC in cropland and grassland soils was distributed in intermediate or stabilised SOC
pools, while this was only true for 60% of the SOC found in forest soils. Therefore, those authors
suggested that Bavarian soils under cropland and grassland are more suitable for long-term
sequestration of additional SOC than soils under forest. They also examined controlling factors for
the SOC distribution among fractions in the different land-uses (Wiesmeier *et al.,* 2014). Correlation
analyses suggested that the intermediate SOC pools in croplands and grasslands were significantly
correlated to soil moisture, but none of the functional SOC pools were influenced by temperature or
precipitation. The particulate organic matter (POM) fraction of soils under grasslands and croplands
was not significantly related to any environmental factor in that study (Wiesmeier *et al.*, 2014).
Poeplau & Don (2013a) conducted a study on 24 sites in Europe and found that SOC fractions
differed in their degree of sensitivity to land-use change (LUC), with the sensitivity declining with
increasing stability in the SOC fractions. Their results indicated that afforestation of cropland shifts
SOC from the more stable to the more labile fractions, while the conversion from cropland to
grassland the newly sequestered SOC is stored in the intermediate to stable pools. Rabbi *et al.* (2014)
examined the relationships between land-use, management, climate and soil properties and the
stock of three SOC fractions for soils in south-eastern Australia, and observed a high impact of
climate and site-specific factors (rainfall, silicon content, soil pH, latitude) and only a minor influence
of land-use. The dominance of site and climate variables as impact factors in that region may
primarily be due to the wide range of site conditions in the area studied.
If the regional distribution of SOC fractions can be predicted using a combination of fractionation
methods and NIRS and if relevant drivers for this distribution can be found, it should be possible to
identify regions in Germany in which soils are most vulnerable to carbon losses. Some carbon
fractions are commonly assumed to be more labile than others because they apparently have lower
turnover times in the soil. The question is if it can simply be assumed that soils that contain a high
percentage of those "labile" fractions are more vulnerable to carbon losses than others. On the one
hand, it should be noted that for the assessment of vulnerability to carbon losses, not only the
distribution of the fractions should play a role, but also the absolute amounts of carbon within the
fractions. This is important as some soils may have stored a high percentage of SOC in a labile form,
but the absolute amount of this SOC may be very low and thus less relevant in terms of climate
change mitigation than a small percentage of light fraction that is lost from a soil rich in SOC. On the
other hand, there are several regions in north-Western Europe and also in northern Germany where
the soils exhibit unusually high SOC content while having a high sand and low clay content (Sleutel *et*
*al.*, 2011). These so called 'black sands' have a poor capacity to stabilise SOC by binding onto mineral
surfaces, and therefore most SOC is present in the form of POM. A great part of this land surface in
northern Germany was covered by heathland and peatland until the end of the 18[th] century and
those soils may behave different than other soils in terms of SOC storage and the vulnerability to
carbon losses may not generally be definable via dividing SOC into fractions by density fractionation.
The present study is part of the German Agricultural Soil Inventory. A set of 145 topsoil samples,
representative of German agricultural soils, was fractionated and used to calibrate NIRS predictions
of the constituent fractions for > 2500 sites with mineral soils all over Germany. Additional climate,
management and geographical data were gathered for all sites and a machine learning algorithm was
employed to clarify which factors influence the distribution of the carbon fractions. In this paper we
therefore aim to answer the following research questions:
1) How is SOC distributed among the fractions at national scale?
2) Which driving factors are relevant for this distribution?
3) Can regions of high vulnerability to carbon losses be identified by this predictive approach?

## 2 Material & Methods

### 2.1 Study area, sampling and sample selection

Germany has a total surface area of 357 000 km² and its climate is temperate, marine and
continental. Mean annual precipitation (MAP) ranges between 490 and 2090 mm and mean annual
temperature (MAT) between 5.7 and 11.2 °C. Around half the country's surface area is used for
agriculture, with cropland accounting for 71% of this area, grassland for 28% and other crops (e.g.
vines) for 1%.
Soil samples were taken in the course of the ongoing German Agricultural Soil Inventory. By May
2017, 2900 agricultural sites (croplands and grasslands) were sampled based on an 8 km x 8 km rid.
At each site, a soil profile was characterised by a soil scientist and soil samples were taken from five
fixed depth increments, using 2-10 sampling rings per depth increment (depending on the stone
content) that were representatively distributed. All soils were classified in the field according to the
German Soil Classification System (Sponagel et al., 2005).
For this study, a representative set of calibration sites was needed to be able to predict the carbon
fractions using NIRS. Therefore, 145 calibration sites were chosen according to the following criteria:
1) Maximum difference in NIR spectra, according to the Kennard-Stone algorithm (Daszykowski et al.,
2002), 2) consistent spatial distribution within Germany, 3) exclusion of sites with SOC content > 87 g
$kg^{-1}$ in any horizon, as such soils may be organic (> 30% organic substance) or in transition between
organic and mineral soils and it was assumed that the processes governing the variability of SOC in
organic soils differ from those in mineral soils, and 4) representative mapping of land-use, soil type
and carbon stocks. The topsoils (0-10 cm) of these 145 sites were fractionated to provide the
calibration set for the prediction of the carbon fractions in the remaining sites using NIRS. After
obtaining the predicted carbon fractions for all 2900 sites, the machine learning algorithm cforest
was employed to identify driving factors important for the distribution of SOC into fractions. The
employed fractionation scheme is described in section 2.3 while details on the NIRS and
chemometrics are given in section 2.4. The use of the cforest algorithm is explained in section 2.5.
*2.2 Laboratory analyses*
All 2900 topsoil samples were dried and analysed for gravimetric water content, electrical
conductivity (EC), pH, SOC content (g $kg^{-1}$, by dry combustion), soil inorganic carbon content (g $kg^{-1}$)
after removing organic carbon in a muffle kiln, texture (by the pipette method), rock content, root
content and bulk density (with repeated soil rings). The SOC stocks were calculated as suggested by
Poeplau *et al.* (2017), taking into account the stone and root content of the soil.

**2.3 Fractionation of calibration samples**

The topsoil samples (0-10 cm depth) of the selected calibration sites were dried at 40°C to constant weight and sieved to a size <2 mm. Three different fractions were prepared, using an adaptation of the fractionation scheme proposed by Golchin et al. (1994):

1) To obtain the fraction that contains intra-aggregate particulate organic matter (iPOM), 20 g of soil sample were placed in a falcon tube, which was then filled to 40 mL with sodium polytungstate (SPT) solution (density=1.8 g mL$^{-1}$). The sample was dispersed ultrasonically at 65 J mL$^{-1}$ to standardize the treatment of the iPOM fraction, which is often isolated by shaking in other studies. The probe energy supply was calibrated using the procedure explained in Puget *et al.* (2000). The tube was centrifuged at 4000 rpm until there was a clear distinction between the iPOM fraction and the remaining soil pellet. The supernatant was then filtered through a 45 μm filter paper and a ceramic filter using vacuum filtration. The iPOM fraction remained on the filter and was rinsed with distilled water until the electrical conductivity of the filtered water was below 10 μS m$^{-1}$. The iPOM fraction was then dried at 40°C, weighed and milled.

2) To obtain the particulate organic matter occluded in aggregates (oPOM) fraction, the falcon tube containing the pellet was again filled to 40 mL with SPT solution. The pellet was mixed with the solution using a vortex shaker and then ultrasonic dispersion was applied again, at 450 J mL$^{-1.}$. This energy level was chosen as Schmidt et al. (1999) found that 450 to 500 J mL$^{-1}$ is enough to disperse all soil aggregates (including microaggregates) in a wide range of soil types. The sample was centrifuged and the oPOM fraction was processed as described above for the iPOM fraction.

3) The remaining soil pellet was assumed to contain the mineral-associated organic matter (MOM or heavy) fraction. The pellet was washed three times with 40 mL of distilled water, dried, weighed and milled in the same way as the iPOM and oPOM fractions. The organic carbon (C) and total nitrogen (N) content of the three fractions were determined through thermal oxidation by dry combustion using an elemental analyser (LECO Corp.). One possible limitation of the applied fractionation scheme is that pyrogenic carbon ends up in the light iPOM and oPOM fractions although it generally has

higher turnover times than assumed for this fraction. For Germany, however, we are confident that
this is not influencing the results, as pyrogenic carbon only plays a minor role in German soils. The
fractionation method applied is only one out of several possible methods and options to separate
labile from stabilised SOC.
The carbon recovery rate of the fractionation approach was between 80 and 110%. Recovery rates of
more than 100% can be reached as the sample that is measured for total SOC and the sample that is
fractionated are not exactly the same. Even through careful subsampling the samples cannot be
completelely homogenized concerning their carbon content. The mean carbon contents of the
fractions were 34.7% for the iPOM fraction, 27.4% for the oPOM fraction and 1.8% for the MOM
fraction.
Basic descriptive statistics were calculated for the data on the fractionated calibration sites, including
mean absolute and relative proportions of the SOC fractions divided between different land-uses and
soil texture classes. An ANOVA was conducted to determine whether the differences between
cropland and grassland land-uses were significant and to test for significant differences between soil
texture classes. The Games Howell post-hoc test was used for this purpose.

**2.4 Near-infrared spectroscopy and chemometrics**
As the oPOM fraction generally contained a small proportion of total SOC (on average 4%), it was not
reliably predictable on its own. Therefore, it was combined with the iPOM fraction to give a 'light
fraction' for the purpose of prediction. This was done even though it is clear that iPOM and oPOM
may differ in their availability for decomposition and in their turnover times. In this case an accurate
prediction of the combined light fraction was thought to be more important and better than an
inaccurate prediction of the oPOM fraction, as this can be misleading for the readers when displayed
on a map.Soil samples were dried at 40°C, sieved through a 2 mm sieve and finely milled in a rotary
mill. Before analysis, the samples were dried again at 40°C and equilibrated to room temperature for
a few minutes in a desiccator. The soil samples were scanned with spot size 4 cm diameter in a
Fourier-Transform near-infrared spectrophotometer (FT-NIRS, MPA - Bruker Optik GmbH, Germany).
Spectral data were measured as absorbance spectra (A) according to A = log (1/R), where R is the
reflectance expressed in wave number from 11000 to 3000 cm$^{-1}$ (NIR region) with 8 cm$^{-1}$ resolution
and 72 scans. The final spectrum was obtained by averaging two replicates.
To improve the model accuracy a spectral pre-treatment was applied, using Savitzky-Golay first
derivative smoothing (3 points) and wavelength selection from 1330 to 3300 nm, since these regions
contain the main absorbance information. The calibration set consisted of the 145 calibration site
samples, and the remaining samples were used for prediction. Partial least squares regression (PLSR)
was performed in the pls package (Mevik et al., 2015), based on near-infrared (NIR) spectra and
reference laboratory data. A cross-validation was applied using leave-one-out to avoid over- and
under-fitting. To obtain the carbon fractions and ensure that the sum of light and heavy fractions was
equal to total SOC content, the log ratio of the light and heavy fraction was predicted (Jaconi et al., in
review). A validation using an independent validation set was not deemed advisable in this study as
the calibration dataset was representative for the whole area of Germany including a diverse set of
soil types and geographical circumstances. Moreover, 145 samples are not a large dataset for a
calibration and with every split of this dataset a large part of the variation present in German soils
would be lost for the calibration. An independent validation using the same dataset was carried out,
however, by Jaconi et al. (in review) and the calibration and validation results can be found in table
S3. Model performance was evaluated using the root mean square error of cross-validation
(RMSECV), Lin's concordance correlation coefficient (ρc) and the coefficient of determination (R²)
between predicted and measured carbon content in the fractions. In addition, the ratio of
performance to inter-quartile range (RPIQ) and the residual prediction deviation (RPD) were
calculated, the latter using the classification system devised by (Chang et al., 2001). This classification
is arbitrary, but nonetheless, can be used to assess the model quality and to compare with other
models.
We used the methodology as above described as Jaconi et al (in review) found that NIRS is one
promising method to predict carbon fractions, which is fast, low-cost and accurate. The authors had
the following calibration results: For prediction of carbon content in the fractions (g kg$^{-1}$), the
coefficient of determination ($R^2$) between predicted and measured carbon content in the fractions
was found to be 0.87-0.90 and RMSECV was 4.37 g kg$^{-1}$. The RPD was 2.9 for the prediction of carbon
content in the light fraction and 3.2 for the prediction in the heavy fraction. For prediction of carbon
proportions in the fractions (%), $R^2$ was 0.83, RMSECV 11.45% and RPD 2.4 (Fig. S1; for more details
see Jaconi *et al.*, in review). The accuracy of prediction of both SOC content and proportions of the
light and heavy SOC fractions was very good and was comparable with that in other studies that have
used NIRS to predict SOC fractions (Cozzolino and Moro, 2006; Reeves et al., 2006).
**2.5 Drivers of soil organic carbon distribution in fractions**
A total of 75 potential drivers of differences in carbon proportions in different fractions was compiled
from the soil analysis data, complemented with data from a farm survey and geographical data (for a
complete list of predictors, see Table S2). The farm survey recorded management practices, over the
10 years, if known by the farmer, prior to sampling. Using this, yearly mean carbon and nitrogen
inputs through plant material and organic and mineral fertilizers were calculated for each site based
on the yield of the main product and on different carbon allocation functions for different crops as
described in (Bolinder et al., 1997)When data were missing in the survey responses, yields were
calculated using regional yield estimates provided by the regional governments. Climate and site data
acquired from GIS data layers completed the set of predictor variables (climate data from Deutscher
Wetterdienst, normalised difference vegetation index (NDVI) data from ESA, elevation data from
Bundesamt für Kartographie und Geodäsie). For the sites in the federal states of Lower-Saxony,
North-Rhine Westphalia, Mecklenburg-Western Pomerania, Rhineland-Palatinate, Saxony Anhalt and
Schleswig Holstein (Northern Germany), the land-use history was researched using historical maps
(dating back to 1873-1909), as many regions in these states are known to have a heathland or
peatland legacy.
The conditional inference forest algorithm (cforest; Hothorn *et al.*, 2006), was used to identify the
most influential drivers of SOC distribution among the different fractions. Cforest is an ensemble
model and uses tree models as base learners that can handle many predictor variables of different
types and can also deal with missing values in the dataset (Elith et al., 2008). The cforest algorithm is
similar to the better known random forest algorithm, a non-parametric data mining algorithm that
uses recursive partitioning of the dataset to find the relationships between predictor and response
variables (Breiman, 2001).
Bootstrap sampling without replacement was carried out in order to prevent biased variable
importance (Strobl et al., 2007). As multicollinearity between the predictors may result in a biased
variable importance measure in cforest algorithms (Nicodemus et al., 2010), the correlations
between the predictor variables were controlled. When the correlation between two possible
predictors was > 0.8, only the one with the broader range of variation was kept in the dataset. Ten
cforest models were created, each containing 1000 trees and using different random subset
generators. From these models, the variable importance of predictors was extracted and the relative
variable importance was calculated and averaged over all 10 models. Variables were considered
important when their relative variable importance was higher than 100/n, where n is the number of
predictors in the model. This is the variable importance that each variable would have in a model
where all variables are equally important (Hobley et al., 2015). It should be noted that the relative
variable importance value obtained from the cforest algorithm does not necessarily imply direct
relationships between the proportion of SOC in the light fraction and the main drivers, as the
algorithm also takes into account interaction effects between the variables. Model performance was
assessed by the coefficient of determination ($R^2$), as defined by the explained variance of out-of-bag
estimates, which represent a validation dataset:

$$R^2 = 1 - \frac{MSE_{OOB}}{Var_z} \qquad (1)$$

where $MSE_{OOB}$ is the mean squared error  of out-of-bag estimates and Var$_z$ is the total variance in
the response variable.
A range of soils in northern Germany, called 'black sands', behaved quite differently from other soils
in the country in terms of the driving factors for SOC distribution among the fractions. Therefore the
dataset was split into two parts for the cforest analysis and the cforest algorithm was used on: 1) the
dataset containing only the black sands from northern Germany (n=264). Those were extracted using
the NIR spectra, which were classified into black sands and normal soils using the simca function in
the "mdatools" package (Kucheryavskiy, 2017); and 2) on all other soils considered not to be black
sands (n=2406). All statistical analyses were conducted using the software R . Maps were generated
with the software QGIS.

## 3 Results

### 3.1 Carbon distribution among measured fractions (145 calibration sites)

The iPOM fraction contributed an average of 23% to bulk SOC (23% ±2.36 (mean ± standard error (SE)) in croplands and 25% ±3.8 in grasslands (Fig. 1). The oPOM fraction accounted for an average of 4% of SOC (3% ± 0.5 in croplands, 8% ±1.3 in grasslands) across all calibration sites (Fig. 1). The heavy fraction contributed the highest proportion to bulk SOC (73% in all soils, 73% ± 2.5 in croplands and 68% ± 4.4 in grasslands). The differences between land-uses were not significant. There was great variation in the carbon distribution between fractions, with the iPOM fraction contributing between 3 and 99% to bulk SOC. The absolute carbon content (g kg$^{-1}$) of the fractions was significantly different for the heavy fraction, with grasslands having significantly higher heavy fraction carbon content than croplands (31 g kg$^{-1}$ ± 3 compared with 13 g kg$^{-1}$ ± 0.7).

There were significant differences in the contribution of the different fractions to bulk SOC depending on the main soil texture class (Fig. 2). In sandy soils, the iPOM fraction contributed significantly more and the heavy fraction contributed significantly less to bulk SOC than in other soils. For the oPOM fraction, the difference between sandy soils and clayey, silty and loamy soils was not significant. The absolute SOC content (g kg$^{-1}$ soil) was significantly higher in the heavy fraction of clayey soils than in the heavy fraction of all other soil textures and it was significantly higher in the oPOM fraction of sandy soils than in the i fraction of all other soils.

### 3.2 Influences on soil organic carbon distribution among fractions (all 2900 sites)

With the machine-learning algorithm cforest, 75 variables that may act as drivers for the regional distribution of SOC fractions were evaluated (Fig. 3a). For the 'normal' soils (non-black sands) dataset, soil texture had the highest explanatory power in predicting the contribution of the light fraction to bulk SOC (Fig. 4), with clay content being negatively and sand content positively correlated with percentage of SOC in the light fractions. The SOC content, bulk soil C/N ratio, land-

use, soil type, pH and $CaCO_3$ content were also identified as important explanatory variables when
predicting the light fraction proportion. The SOC content showed a positive relationship with light-
fraction SOC proportion and with bulk soil C/N ratio. The grassland soils showed a higher proportion
of bulk SOC in the light fraction than the cropland soils and pH was negatively related to the light-
fraction SOC proportion. Comparing the fractions distribution in the different soil types, it is obvious
that podzols store a substantially higher proportion of their total SOC in the light fraction than all
other soil types (Fig. 6).
The analysis of historical land-use data of northern Germany confirmed that the former peatland,
heathland and grassland sites had significantly higher ($p < 0.01$) proportions of bulk SOC in the light
fraction than sites used as cropland in the same period (Fig. 5a). These historical peatland, heathland
and forest sites also had significantly higher ($p<0.05$) C/N ratio than the historical cropland and
grassland sites (Fig. 5b). Regarding the total SOC content, historical peatland and grassland sites had
significantly higher ($p<0.001$) values than historical croplands (Fig. 5c).
For the black sands dataset, bulk soil SOC content was the most important driver of SOC distribution
in the fractions (Fig. 3b), followed by C/N ratio, soil temperature in summer and soil bulk density. The
SOC content had a positive relationship with percentage of SOC in the light fraction, and with C/N
ratio (Fig. 4). For soil temperature there was no clear relationship. There was a negative relationship
between SOC proportion in the light fraction and soil bulk density.
**3.3 Distribution of soil organic carbon into fractions across Germany**
Regions featuring high proportions of SOC in the light fraction (over 60% of total SOC) nearly all lie in
northern Germany (Fig. 7). Medium proportions of SOC in the light fraction (40-60% of total SOC)
were found in Mecklenburg-Western Pomerania and in parts of Brandenburg (north-east Germany).
Low proportions (< 40 %) of SOC in the light fraction were found in central and southern Germany.
Considering the absolute contents of SOC in the light fraction (Fig. 8), it was obvious that the
absolute (in g/kg) and relative (in %) carbon contents in the light fraction are in close alignment in
most regions in Germany, implying that those sites with a higher total SOC content also have a higher
proportion of this content stored in the light fraction.

## 4 Discussion

### 4.1 Contribution of soil organic carbon fractions to bulk soil organic carbon

The relative distribution of carbon among different fractions did not differ significantly between croplands and grasslands (Fig. 2a) in the calibration dataset (n=145) which is in agreement with previous findings for south-east Germany (Wiesmeier et al., 2014). There was a trend, however, for slightly higher iPOM content in grasslands than in croplands. When taking the full dataset, including the fractions predicted with NIRS, the difference was significant ($p < 0.05$), with higher proportions of POM in grassland topsoils when compared to cropland (not shown). Other studies, however, found considerably higher differences between POM proportions in grassland and cropland soils. Christensen (2001) estimated that, in grassland soils, 15-40% of SOC is stored in the light fraction and Poeplau and Don (2013b) found the light fraction proportion to be twice as high in grassland topsoils (0-10 cm) compared to cropland soils. One possible reason for a larger light fraction in grassland soils is the permanent vegetation cover and the high amount of roots, which provide a higher aboveground and belowground input of SOC (Christensen, 2001). The limited differencein light fraction between cropland and grassland soils shown in our study can possibly be due to interfering factors, as historical land-use changes which would need deeper investigations to unravel. Moreover, grasslands and croplands are generally located on different soil types which, again, interferes with other factors as soil moisture or texture. Therefore, it is not always possible to draw direct conclusions on land-use change effects on carbon fractions from such regional inventories.

The significant differences observed in the absolute SOC content of fractions between different land-uses were to be expected, as grassland soils in Germany contain on average more than twice as much SOC in the upper 10 cm as cropland soils ($42\pm16$ g kg$^{-1}$ compared with $17\pm9$ g kg$^{-1}$, Fig. 2b). This higher carbon content of grassland soils is often found and can mainly be attributed to the higher SOC inputs and the lack of tillage induced SOC mineralization in the topsoil (Post and Kwon, 2000; Wiesmeier et al., 2014).

All samples with medium or high proportions of SOC in the light fraction were found to originate from northern Germany. This is the area in which the black sands are present, which store large parts of their SOC in the light fraction. Springob & Kirchmann (2002a) examined the presence of black sands in Lower Saxony in Germany and linked it to the land-use history. In Ap-horizons of soils formerly used as heathland or plaggen, they found a high fraction of SOC resistant to oxidation with HCl. This HCl-resistant fraction was positively correlated with the total SOC content, but soil microbial biomass carbon content showed a negative relationship with total SOC and, when incubated, the specific respiration rates were lowest for the soils with the highest SOC content (Springob & Kirchmann, 2002a). Those authors concluded that a high proportion of the organic matter in the former heathland soils is resistant to decomposition and suggested that low solubility of the SOC could be responsible for its high stability. A recent study (Alcántara et al., 2016) reported similar results for sandy soils under former heathland, which had lower respiration rates per unit SOC and a wider range of C/N ratios than control soils without a heathland history. Certini *et al.* (2015) showed that SOC under heathlands is rich in alkyl C and contains high contents of lipids, waxes, resins and suberin, all of which hinder microbial degradation. This confirms the claim that sandy soils under former heathland and contain high contents of stable SOC even though they also contain a high amount of POM. In such soils, the POM fractions may not be directly linked to higher turnover rates and lower stability.

"Historical" peatlands may have lost much of their former carbon stocks due to a number of reasons: Drained peatlands emit huge amounts of $CO_2$ (German grasslands on average 27.7 to $CO_2$ ha$^{-1}$ yr$^{-1}$, (Tiemeyer et al., 2016)) until the peat has virtually vanished. There might have also been peat extraction, and the remaining peat layer might have been mixed with underlying sand. Finally, former peatland soils were often mixed with large amounts of sand in order to make them usable for arable cultivation, but still often contain substantial proportions of (degraded) peat and therefore have relatively high SOC content, with a large part of the SOC in the light fraction. It has been found elsewhere (Bambalov, 1999; Ross and Malcolm, 1988; Zaidelman and Shvarov, 2000) that the SOC

content in sand-mix cultures declines rapidly after mixing with sand and that the decline increases
with increasing intensity of mixing. In a 15-year long-term trial, Bambalov (1999) found that the SOC
content of a sand-mix culture could only be stabilised (at much lower SOC content than the original
peat) by adding organic and mineral fertilisers to the soil. In contrast, Leiber-Sauheitl et al. (2014)
found that a peat-sand mixture with a SOC content of 93 g kg$^{-1}$ emitted as much $CO_2$ as an adjacent
shallow "true" peat. Similarly, Frank *et al.* (2017) determined a higher contribution of soil-derived
dissolved organic carbon at a peat-sand mixture compared to the peat, which points to a low stability
of the SOC in this kind of soils. This means that, for the light fraction of the former peatlands in
northern Germany, enhanced stability of the POM cannot be assumed. Thus, for more accurate
interpretation of results, the black sands had to be divided into a former heathland group, containing
a relatively stable light fraction, and a former peatland group, containing a relatively labile light
fraction, although there are transitional vegetation types with heath on peatlands.
Land-use history clearly continues to influence soil SOC dynamics, since the light-fraction SOC
proportion and the bulk soil C/N ratio were higher in soils with a heathland or peatland history in the
present study. This supports findings by Sleutel *et al.* (2008) that the chemical composition of pairs
of relict heathland and cultivated former heathland soils is very similar. Unfortunately former
peatlands and heathlands are not necessarily distinguishable due to their SOC content and C/N ratio,
so that knowledge on the land-use history is necessary. In some cases, however, even the distinction
on site can be difficult, e.g. on dry peatlands with heath vegetation (*Calluna, Erica*). In future studies
it would therefore be interesting to incubate pairs of former heathland and peatland in order to be
able to make accurate claims on the vulnerability of the light fraction SOC in these soils.
The presence of black sands poses a problem for interpretation of the SOC fractions. In most cases,
the SOC in the light fraction (iPOM + oPOM fractions) is seen as representing a labile carbon pool
with short turnover times. Therefore sites with high proportions of bulk SOC in the light fraction
would be seen as being at risk of losing this substantial part of their SOC stock quite rapidly and
easily. For the black sands, however, their former heathland land-use history has led to quite stable
and not easily degradable POM (Overesch, 2007; Sleutel et al., 2008; Springob and Kirchmann, 2002),
while for former peatland that was drained and possibly mixed with sand the classification of the
light fraction into a labile SOC pool may well be justified (Leiber-Sauheitl et al., 2014). This implies
that the results need to be interpreted in a different way for black sands than for other soils.
**4.2 Driving factors for carbon distribution into fractions**
**4.2.1 'Normal' agricultural soils (non-black sands)**
The most important driver for the SOC distribution among the fractions in 'normal' soils was the soil
texture (Fig. 3a). This is well in line with the frequently reported relationship between clay content
and mineral-associated (heavy fraction) SOC, whereby clayey soils can stabilise SOC through
mechanisms that protect it against microbial decay by absorption or occlusion (v. Lützow et al., 2006;
Six et al., 2002). The SOC that is bound to the mineral phase is mostly assigned to a conceptual stable
SOC pool. The negative relationship between SOC content and percentage of SOC in the heavy
fraction (Fig. 4) may indicate SOC saturation of the mineral fraction at rising SOC content, so that
excess SOC can only be stored as particulate organic carbon.
The positive correlation between soil C/N ratio and C proportion in the light fraction (Fig. 4) is related
to the inherent higher C/N ratio of the light fraction compared with the heavy fraction. Thus, a higher
share of light-fraction C leads to a higher C/N ratio of the bulk soil. Thus, in 'normal' agricultural soils
the C/N ratio may be useful as an indicator of SOC stability: A high C/N ratio indicates a high
proportion of labile SOC in the soil.
The fact that land-use is an important driver for the distribution of SOC among the fractions is mainly
due to the fact that in the dataset containing all non-black sand sites topsoils under grassland store a
significantly higher share of SOC in the light fraction than topsoils under cropland. This is in line with
higher inputs of roots, which make up part of the light fraction, into grassland topsoils. The higher
proportion of SOC in the light fraction was also noted in the calibration dataset (n=145), but the
difference was not significant in that case.
Apart from texture, C/N ratio and land-use, another important driving factor for the distribution of
SOC among fractions was the soils carbonate content. Most arable topsoils in Germany do not
contain carbonate. The 9% of arable soils that contained over 5% carbonate in this study consistently
had a high proportion of heavy-fraction carbon and were therefore classified as containing mainly
stabilised SOC (Fig. 4). Calcium bridges may foster absorption of SOC onto mineral surfaces and, via
an active soil fauna, high pH enhances the turnover and transformation of SOC from recently added
biomass to mineral-associated SOC that can be stabilised via absorption (Oades, 1984). In general,
there was a trend for a higher proportion of SOC in the light fraction with lower pH (Fig. 4), which is
well in line with the finding by Rousk *et al.* (2009) that SOC mineralisation is slower in soils with lower
pH due to a higher ratio of fungal to bacterial biomass.
The influence of soil type is mainly due to the Podzol soils storing a much higher proportion of bulk
SOC in the light fraction than all other soil type classes (Fig. 6). Podzols often develop on sandy soils
and therefore do not have a high capacity for SOC stabilisation in the heavy fraction (Sauer et al.,

467     2007).

**4.2.2 Black sands**
In the dataset containing only the black sands, soil total SOC content was the most important driver
for the SOC distribution among the fractions, with increasing light fraction with increasing SOC
content (Fig. 4). On the one hand, this could indicate saturation of the heavy fraction at high SOC
contents, which would lead to further storage in the light fraction only, as already mentioned above
for 'normal' soils. Another possible explanation is that those soils with the highest SOC content in the
dataset are degraded peatlands, in which a high percentage of the SOC ends up in the light fraction.
On former heathlands, the soil total SOC content is also quite high compared with that in other sandy
soils and the light fraction is mainly built up from *Calluna vulgaris* litter, since *Calluna* vegetation
dominates on many heathlands. *Calluna* litter contains very stable SOC due to high contents of lipids,
long-chain aliphatics and sterols, and may persist in the light fraction of soil for decades or even
centuries (Sleutel et al., 2008).
There is a close link between land-use history as peatland and heathland and soil C/N ratio, with high
C/N ratio in former heathland soils (Alcántara et al., 2016; Certini et al., 2015; Rowe et al., 2006) and
also often in former peatlands (Aitkenhead and Mcdowell, 2000). Therefore it is evident that land-
use history is a main driver for the high proportions of bulk SOC found in the light fraction in these
soils. This is well in line with the significantly higher C/N ratios reported for soils in Lower-Saxony and
Mecklenburg-Western Pomerania, which were under heathland or peatland more than 100 years ago
(Fig. 5). The influence of land-use history reinforces the relationship between C/N ratio and the light
fraction.
In black sands, there was a significant negative relationship between soil temperature and the light-
fraction SOC proportion, but this was not found for the other soils (Fig. 4). A negative relationship
was observed between soil bulk density and proportion of SOC in the light fraction, which was
evidently due to the low density of the light fraction affecting overall soil bulk density (Fig. 4).
Even though the land-use history was part of the dataset and we could link several of the important
driving factors to a history as peatland or heathland, the cforest algorithm did not identify the land-
use history as important driver for the SOC distribution into fractions. This was the case because we
did not have the detailed land-use history data for all sites. But even when running the cforest
algorithm only for those sites with known land-use history, it was not selected as important driver.
This is probably due to the fact that at the time of the land survey in 1873-1909 some of the former
heathland and peatland sites had already been cultivated. Therefore the land-use history would not
prove as a reliable indicator. We did confirm this by referring to an older land survey, dating back to
1764-1785. For sites that exhibited typical black sand features (e.g. high SOC proportions in light
fractions, high sand content, and high C/N ratio) but were not a heathland and peatland in the 19[th]
century, we often found a heathland or peatland signature on the maps from the 18[th] century.
Unfortunately this land survey from the 18[th] century is incomplete and we could therefore not rely
on it for all sites.
**4.3 Hot regions of labile and stable carbon in Germany**
land-useFor a soil to be definitively identified as being vulnerable to SOC losses, it not only needs to
have a high proportion of bulk SOC in the light fraction, but also a high absolute SOC content in this
fraction.  The map in Fig. 8 shows the absolute SOC content of the light fraction at sites of the
German Agricultural Soil Inventory. Comparing Fig. 7 and Fig. 8, it is evident that sites which store a
high proportion of their SOC in the light fraction generally also have high absolute SOC content in the
light fraction. This implies that those sites are really the most vulnerable to SOC losses, as they not
only have high proportions of SOC in the light fraction, but also the highest absolute SOC content in
the light fractions to lose. As the SOC in former peatland soils has been shown to be easily
mineralised (Bambalov, 1999), management of such sites should be aimed at stabilising the SOC
stocks and preventing further degradation of the peat. When there is a heathland history, it can be
assumed that the SOC in the light fraction is quite stable, but that does not imply that freshly added
litter will also be stable. In fact, it is quite likely that it will not be stable if no heathland vegetation is
planted. This implies that the SOC stocks on these sites will decline when the resistant litter is not
replenished.
Taking together all the important explanatory variables discussed above, regions in which the SOC
can be classified as mostly labile were identified. These were soils with a high proportion of light
fraction and without a heathland history. Such soils are mainly located in northern Germany and
many of those have a peatland history (Fig. 7). These soils can be seen as vulnerable to losses of a
high proportion of their SOC in the topsoil easily and rapidly. Loss of SOC could occur e.g. through a
change in management that reduces carbon inputs to the soil and therefore fails to maintain the light
fraction, for example a land-use change from grassland to cropland (Poeplau et al., 2011) or reduced
input of organic fertilisers or crop residues (Dalal et al., 2011; Srinivasarao et al., 2014). Losses of SOC
could also occur due to higher temperatures, which could lead to enhanced microbial activity and
therefore enhanced mineralisation of SOC in the light fraction (e.g. Knorr *et al.*, 2005). Former
peatland soils may already lose significant parts of their SOC (Leiber-Sauheitl et al., 2014; Tiemeyer et
al., 2016).
Regions with soils with a high proportion of stable SOC are located mainly in central and southern
Germany (Fig. 7). In these regions, soils consistently store over 60% of their SOC in the heavy
fraction, in which the SOC is bound mostly to the mineral surfaces of clay minerals. Thus, these soils
have the lowest vulnerability to losing their SOC, as losses mostly occur from the light fraction.
However, even in these regions up to 40% of bulk SOC is stored in the light fraction and this may be
lost. Therefore apparent lower vulnerability does not mean that SOC-conserving soil management is
not needed in these regions. It should be noted that the quality of the SOC in the light fraction is
probably not the same in all soils, land-use (history) and climate regions. Therefore, the vulnerability
and turnover time of the light fraction may also vary considerably within different regions.  This can
be seen in the light fraction C/N ratio for example, which ranged between 11 and 43 for the 143
calibration sites studied here.
Using the combination of SOC fractionation and prediction with NIRS, it is generally possible to
identify regions that are more or less vulnerable to SOC losses. The results must be assessed with
care, however, as phenomena like non-labile light fraction in black sands can hamper the
interpretation. It is therefore advisable to look at different driving factors when classifying sites as
more vulnerable than others. Moreover, special soil phenomena are to be assessed separately from
'normal' soils, as the driving factors for the fractions distribution may vary considerably.

## 5 Conclusions

Identification of the distribution of SOC fractions in German soils allowed clear identification of regions where the SOC in agricultural soils is most vulnerable to being lost. The cforest analysis provided indications of the factors driving the distribution of SOC into the different fractions. It was found that soil texture, bulk soil SOC content, bulk soil C/N ratio, land-use history and pH were the main drivers for this distribution in 'normal' soils. In 'black sand' soils in northern Germany, the SOC distribution into the fractions mainly depended on total SOC content and soil C/N ratio and was directly linked to the land-use history. Former peatland or heathland still has a great influence on the composition of soil SOC decades or even centuries after cultivation of the soil. In some regions of Germany the majority of bulk SOC is stored in the light fraction, but this does not always imply that this SOC is labile. Use of SOC fractionation techniques coupled with NIR spectroscopy to extrapolate to a national soil inventory dataset was successful in predicting POM factions. However, additional knowledge on land-use history was required to determine whether this POM is vulnerable to losses or not.  This study focused on the topsoil only, as it has comparatively high SOC stocks and is most vulnerable to changes in management. Future studies should also examine the SOC distribution in the subsoil, as this would enable exploitation of all possibilities for sequestering additional SOC in the soil, in order to mitigate the $CO_2$ content in the atmosphere. Regarding soil management measures, this study provided indications on where the most prudent and SOC-conserving management techniques are advisable for different regions of Germany: former peatland soils in Northern Germany are most vulnerable and former heathland soils in the same region are less vulnerable at the moment. The vulnerability of those heathland soils can change, however, when changes in soil management occur. This study showed that through the spatial upscaling of SOC fraction distribution through NIRS prediction, it is possible to elucidate the SOC vulnerability and driving factors for SOC stability on a national scale.

574

575

## Acknowledgements

This study was funded by the German Federal Ministry of Food and Agriculture in the framework of the German Agricultural Soil Inventory. We thank the field and laboratory teams of the German Agricultural Soil Inventory for their thorough and persistent work with the soil samples. Special thanks go to Anita Bauer for her support with the SOC fractionation. We also want to thank Catharina Riggers, Florian Schneider and Christopher Poeplau for valuable comments and discussion of a former version of this manuscript. We thank Norbert Bischoff, Jochen Franz, Andreas Laggner, Lena Liebert, and Johanna Schröder. Our thanks also go to the Bundesamt für Kartographie und Geodäsie and the Deutscher Wetterdienst for providing geodata and climate data, respectively and to the Landesamt für Geoinformation und Landesvermessung Niedersachsen and the Landesamt für innere Verwaltung - Koordinierungsstelle für Geoinformationswesen for providing data on historical land-use.

## Literature

Aitkenhead, J. A. and Mcdowell, W. H.: Soil C : N ratio as a predictor of annual riverine DOC flux at local and global scales, Global Biogeochem. Cycles, 14(1), 127–138, 2000.

Alcántara, V., Don, A., Well, R. and Nieder, R.: Deep ploughing increases agricultural soil organic matter stocks, Glob. Chang. Biol., 22(8), 2939–2956, doi:10.1111/gcb.13289, 2016.

Baker, J. M., Ochsner, T. E., Venterea, R. T. and Griffis, T. J.: Tillage and soil carbon sequestration-What do we really know?, Agric. Ecosyst. Environ., 118(1–4), 1–5, doi:10.1016/j.agee.2006.05.014, 2007.

Baldock, J. A., Hawke, B., Sanderman, J. and Macdonald, L. M.: Predicting contents of carbon and its component fractions in Australian soils from diffuse re fl ectance mid-infrared spectra, Soil Res., 51, 577–595, 2013.

Bambalov, N.: Dynamics of organic matter in peat soil under the conditions of sand-mix culture during 15 years, Int. Agrophysics, 13, 269–272, 1999.

Bolinder, M. A., Angers, D. A. and Dubuc, J. P.: Estimating shoot to root ratios and annual carbon inputs in soils for cereal crops, Agric. Ecosyst. Environ., 63(1), 61–66, doi:10.1016/S0167-8809(96)01121-8, 1997.

Breiman, L.: Random forests, Mach. Learn., 45(1), 5–32, doi:10.1023/A:1010933404324, 2001.

Certini, G., Vestgarden, L. S., Forte, C. and Strand, L. T.: Litter decomposition rate and soil organic matter quality in a patchwork heathland of southern Norway, SOIL, 1, 207–216, doi:10.5194/soil-1-207-2015, 2015.

Chan, K. Y., Heenan, D. P. and Oates, A.: Soil carbon fractions and relationship to soil quality under different tillage and stubble management, Soil Tillage Res., 63, 133–139, 2002.

Chang, C., Laird, D. and Mausbach, M. J.: Near-Infrared Reflectance Spectroscopy – Principal Components Regression Analyses of Soil Properties, Soil Sci. Soc. Am. J., 65, 480–490, doi:10.2136/sssaj2001.652480x.Rights, 2001.

Chimento, C., Almagro, M. and Amaducci, S.: Carbon sequestration potential in perennial bioenergy crops : the importance of organic matter inputs and its physical protection, Glob. Chang. Biol. Bioenergy, 8, 111–121, doi:10.1111/gcbb.12232, 2016.

Christensen, B. T.: Physical fractionation of soil and structural and functional complexity in organic matter turnover, Eur. J. Soil Sci., 52(3), 345–353, doi:10.1046/j.1365-2389.2001.00417.x, 2001.

Cozzolino, D. and Moro, A.: Potential of near-infrared reflectance spectroscopy and chemometrics to predict soil organic carbon fractions, Soil Tillage Res., 85, 78–85, doi:10.1016/j.still.2004.12.006, 2006.

Dalal, R. C., Allen, D. E., Wang, W. J., Reeves, S. and Gibson, I.: Organic carbon and total nitrogen stocks in a Vertisol following 40 years of no-tillage, crop residue retention and nitrogen fertilisation, Soil Tillage Res., 112(2), 133–139, doi:10.1016/j.still.2010.12.006, 2011.

Daszykowski, M., Walczak, B. and Massart, D. L.: Representative subset selection, Anal. Chim. Acta, 468(March), 91–103, 2002.

Devine, S., Markewitz, D., Hendrix, P. and Coleman, D.: Soil Aggregates and Associated Organic

Matter under Conventional Tillage , No-Tillage , and Forest Succession after Three Decades, PLoS
One, 9(1), 1–12, doi:10.1371/journal.pone.0084988, 2014.
Elith, J., Leathwick, J. R. and Hastie, T.: A working guide to boosted regression trees., J. Anim. Ecol.,
77(4), 802–13, doi:10.1111/j.1365-2656.2008.01390.x, 2008.
Frank, S., Tiemeyer, B., Bechtold, M., Lücke, A. and Bol, R.: Effect of past peat cultivation practices on
present dynamics of dissolved organic carbon, Sci. Total Environ., 574, 1243–1253,
doi:10.1016/j.scitotenv.2016.07.121, 2017.
Golchin, A., Oades, J. M., Skjemstad, J. O. and Clarke, P.: Study of Free and Occluded Particulate
Organic Matter in Soils by Solid state 13C CP/MAS NMR Spectroscopy and Scanning Electron
Microscopy, Aust. J. Soil Res., 32, 285–309, 1994.
Helfrich, M., Ludwig, B., Buurman, P. and Flessa, H.: Effect of land use on the composition of soil
organic matter in density and aggregate fractions as revealed by solid-state 13C NMR spectroscopy,
Geoderma, 136(1–2), 331–341, doi:10.1016/j.geoderma.2006.03.048, 2006.
Hothorn, T., Hornik, K. and Zeileis, A.: Unbiased Recursive Partitioning : A Conditional Inference
Framework Unbiased Recursive Partitioning :, J. Comput. Graph. Stat. Comput. Graph. Stat., 15(3),
651–674, 2006.
Jaconi, A., Poeplau, C., Ramirez-Lopez, L., van Wesemael, B., Don, A.: Log-ratio transformation is the
key to determining soil organic carbon fractions using near-infrared spectroscopy.  In review,
European Journal of Soil Science.
John, B., Yamashita, T., Ludwig, B. and Flessa, H.: Storage of organic carbon in aggregate and density
fractions of silty soils under different types of land use, Geoderma, 128, 63–79,
doi:10.1016/j.geoderma.2004.12.013, 2005.
Knorr, W., Prentice, I. C., House, J. I. and Holland, E. A.: Long-term sensitivity of soil carbon turnover
to warming, Nature, 433(January), 298–301, doi:10.129/2002PA000837, 2005.
Lee, J., Hopmans, J. W., Rolston, D. E., Baer, S. G. and Six, J.: Determining soil carbon stock changes:
Simple bulk density corrections fail, Agric. Ecosyst. Environ., 134(3–4), 251–256,
doi:10.1016/j.agee.2009.07.006, 2009.
Lehmann, J. and Kleber, M.: The contentious nature of soil organic matter, Nature, 528, 0–8,
doi:10.1038/nature16069, 2015.
Leiber-Sauheitl, K., Fuß, R., Voigt, C. and Freibauer, A.: High CO2 fluxes from grassland on histic
gleysol along soil carbon and drainage gradients, Biogeosciences, 11(3), 749–761, doi:10.5194/bg-11-
658    749-2014, 2014.

Liu, E., Ghirmai, S., Yan, C., Yu, J., Gu, R., Liu, S., He, W. and Liu, Q.: Long-term effects of no-tillage
management practice on soil organic carbon and its fractions in the northern China, Geoderma, 213,
379–384, doi:10.1016/j.geoderma.2013.08.021, 2014.
Luo, Z., Wang, E. and Sun, O. J.: Can no-tillage stimulate carbon sequestration in agricultural soils? A
meta-analysis of paired experiments, Agric. Ecosyst. Environ., 139(1–2), 224–231,
doi:10.1016/j.agee.2010.08.006, 2010.
v. Lützow, M., Kögel-Knabner, I., Ekschmitt, K., Matzner, E., Guggenberger, G., Marschner, B. and
Flessa, H.: Stabilization of organic matter in temperate soils : mechanisms and their relevance under
different soil conditions - a review, Eur. J. Soil Sci., 57, 426–445, doi:10.1111/j.1365-
2389.2006.00809.x, 2006.
Nicodemus, K. K., Malley, J. D., Strobl, C. and Ziegler, A.: The behaviour of random forest
permutation-based variable importance measures under predictor correlation., BMC Bioinformatics,
11, 110, doi:10.1186/1471-2105-11-110, 2010.
Oades, J. M.: Soil organic matter and structural stability: mechanisms and implications for
management, Plant Soil, 76(1–3), 319–337, doi:10.1007/BF02205590, 1984.
Overesch, M.: Kohlenstoff- und Stickstoffumsatz in Sandböden Niedersachsens, Hochschule Vechta.,
675  2007.

Poeplau, C. and Don, A.: Sensitivity of soil organic carbon stocks and fractions to different land-use
changes across Europe, Geoderma, 192(1), 189–201, doi:10.1016/j.geoderma.2012.08.003, 2013a.
Poeplau, C. and Don, A.: Sensitivity of soil organic carbon stocks and fractions to different land-use
changes across Europe, Geoderma, 192, 189–201, doi:10.1016/j.geoderma.2012.08.003, 2013b.
Poeplau, C., Don, A., Vesterdal, L., Leifeld, J., Van Wesemael, B., Schumacher, J. and Gensior, A.:
Temporal dynamics of soil organic carbon after land-use change in the temperate zone - carbon
response functions as a model approach, Glob. Chang. Biol., 17(7), 2415–2427, doi:10.1111/j.1365-
2486.2011.02408.x, 2011.
Poeplau, C., Vos, C. and Don, A.: Soil organic carbon stocks are systematically overestimated by
misuse of the parameters bulk density and rock fragment content, SOIL, 3, 61–66, doi:10.5194/soil-3-
686  61-2017, 2017.

Post, M. and Kwon, K. C.: Soil Carbon Sequestration and Land-Use Change : Processes and Potential,
Glob. Chang. Biol., 6, 317–328, 2000.
Puget, P., Chenu, C. and Balesdent, J.: Dynamics of soil organic matter associated with particle-size
fractions of water-stable aggregates, Eur. J. Soil Sci., 51, 595–605, 2000.
Rabbi, S. M. F., Tighe, M., Cowie, A., Wilson, B. R., Schwenke, G., Mcleod, M., Badgery, W. and
Baldock, J.: The relationships between land uses , soil management practices , and soil carbon
fractions in South Eastern Australia, Agric. , Ecosyst. Environ., 197, 41–52,
doi:10.1016/j.agee.2014.06.020, 2014.
Reeves, J. B., Follett, R. F., Mccarty, G. W., Kimble, J. M., Reeves, J. B., Follett, R. F., Mccarty, G. W.
and John, M.: Can Near or Mid - Infrared Diffuse Reflectance Spectroscopy Be Used to Determine Soil
Carbon Pools ?, Commun. Soil Sci. Plant Anal., 37, 2307–2325, doi:10.1080/00103620600819461,
698  2006.

Ross, S. M. and Malcolm, D. C.: Modelling nutrient mobilisation in intensively mixed peaty heathland
soil, Pla, 121, 113–121, 1988.
Rousk, J., Brookes, P. C. and Bååth, E.: Contrasting Soil pH Effects on Fungal and Bacterial Growth
Suggest Functional Redundancy in Carbon Mineralization, Appl. Environ. Microbiol., 75(6), 1589–
1596, doi:10.1128/AEM.02775-08, 2009.
Rowe, E. C., Evans, C. D., Emmett, B. A., Reynolds, B., Helliwell, R. C., Coull, M. C. and Curtis, C. J.:
Vegetation Type affects the Relationship between Soil Carbon to Nitrogen Ratio and Nitrogen
Leaching, Water. Air. Soil Pollut., 177, 335–347, doi:10.1007/s11270-006-9177-z, 2006.
Sauer, D., Sponagel, H., Sommer, M., Giani, L., Jahn, R. and Stahr, K.: Podzol: Soil of the year 2007. A
review on its genesis, occurrence, and functions, J. Plant Nutr. Soil Sci., 170(5), 581–597,
doi:10.1002/jpln.200700135, 2007.

Schmidt, M. W. I., Rumpel, C. and Ko, I.: Evaluation of an ultrasonic dispersion procedure to isolate primary organomineral complexes from soils, , (March), 87–94, 1999.

Six, J., Conant, R. T., Paul, E. a and Paustian, K.: Stabilization mechanisms of soil organic matter: Implications for C-saturatin of soils, Plant Soil, 241, 155–176, doi:10.1023/A:1016125726789, 2002.

Sleutel, S., Leinweber, P., Ara Begum, S., Kader, M. A., Van Oostveldt, P. and Neve, S. De: Composition of organic matter in sandy relict and cultivated heathlands as examined by pyrolysis-field ionization MS, Biogeochemistry, 89, 253–271, doi:10.1007/s10533-008-9217-4, 2008.

Sleutel, S., Leinweber, P., Van Ranst, E., Kader, M. A. and Jegajeevagan, K.: Organic Matter in Clay density Fractions from Sandy Cropland Soils with Differing Land-Use History, Soil Sci. Soc. Am. J., 75(2), 521–532, 2011.

Sponagel, H., Grottenthaler, W., Hartmann, K. J., Hartwich, R., Janetzko, P., Joisten, H., Kühn, D., Sabel, K. J. and Traidl, R., Eds.: Bodenkundliche Kartieranleitung (German manual of soil mapping, KA5), 5th ed., Bundesanstalt für Geowissenschaften und Rohstoffe, Hannover., 2005.

Springob, G. and Kirchmann, H.: C-rich sandy Ap horizons of specific historical land-use contain large fractions of refractory organic matter, Soil Biol. Biochem., 34(11), 1571–1581, doi:10.1016/S0038-0717(02)00127-X, 2002.

Srinivasarao, C. H., Venkateswarlu, B., Lal, R., Singh, A. K., Kundu, S., Vittal, K. P. R., Patel, J. J. and Patel, M. M.: Long-Term Manuring and Fertilizer Effects on Depletion of Soil Organic Carbon Stocks Under Pearl Millet-Cluster Bean-Castor Rotation in Western India, L. Degrad. Dev., 25(2), 173–183, doi:10.1002/ldr.1158, 2014.

Strobl, C., Boulesteix, A.-L., Zeileis, A. and Hothorn, T.: Bias in random forest variable importance measures: illustrations, sources and a solution., BMC Bioinformatics, 8, 25, doi:10.1186/1471-2105-8-25, 2007.

Tiemeyer, B., Albiac Borraz, E., Augustin, J., Bechtold, M., Beetz, S., Beyer, C., Drösler, M., Ebli, M., Eickenscheidt, T., Fiedler, S., Förster, C., Freibauer, A., Giebels, M., Glatzel, S., Heinichen, J., Hoffmann, M., Höper, H., Jurasinski, G., Leiber-Sauheitl, K., Peichl-Brak, M., Roßkopf, N., Sommer, M. and Zeitz, J.: High emissions of greenhouse gases from grasslands on peat and other organic soils, Glob. Chang. Biol., 22(12), 4134–4149, doi:10.1111/gcb.13303, 2016.

Wiesmeier, M., Schad, P., Lützow, M. Von, Poeplau, C., Spörlein, P., Geuß, U., Hangen, E., Reischl, A., Schilling, B. and Kögel-knabner, I.: Quantification of functional soil organic carbon pools for major soil units and land uses in southeast Germany ( Bavaria ), "Agriculture, Ecosyst. Environ., 185, 208–220, doi:10.1016/j.agee.2013.12.028, 2014.

Zaidelman, F. R. and Shvarov, A. P.: Hydrothermic regime , dynamics of organic matter and nitrogen in drained peaty soils at different sanding modes, Arch, 45(2), 123–142, doi:10.1080/03650340009366117, 2000.

Zimmermann, M., Leifeld, J., Schmidt, M. W. I., Smith, P. and Fuhrer, J.: Measured soil organic matter fractions can be related to pools in the RothC model, Eur. J. Soil Sci., 58(3), 658–667, doi:10.1111/j.1365-2389.2006.00855.x, 2007a.

Zimmermann, M. Ã., Leifeld, J. and Fuhrer, J.: Quantifying soil organic carbon fractions by infrared-spectroscopy, Soil Biol. Biochem., 39, 224–231, doi:10.1016/j.soilbio.2006.07.010, 2007b.

**Figures**

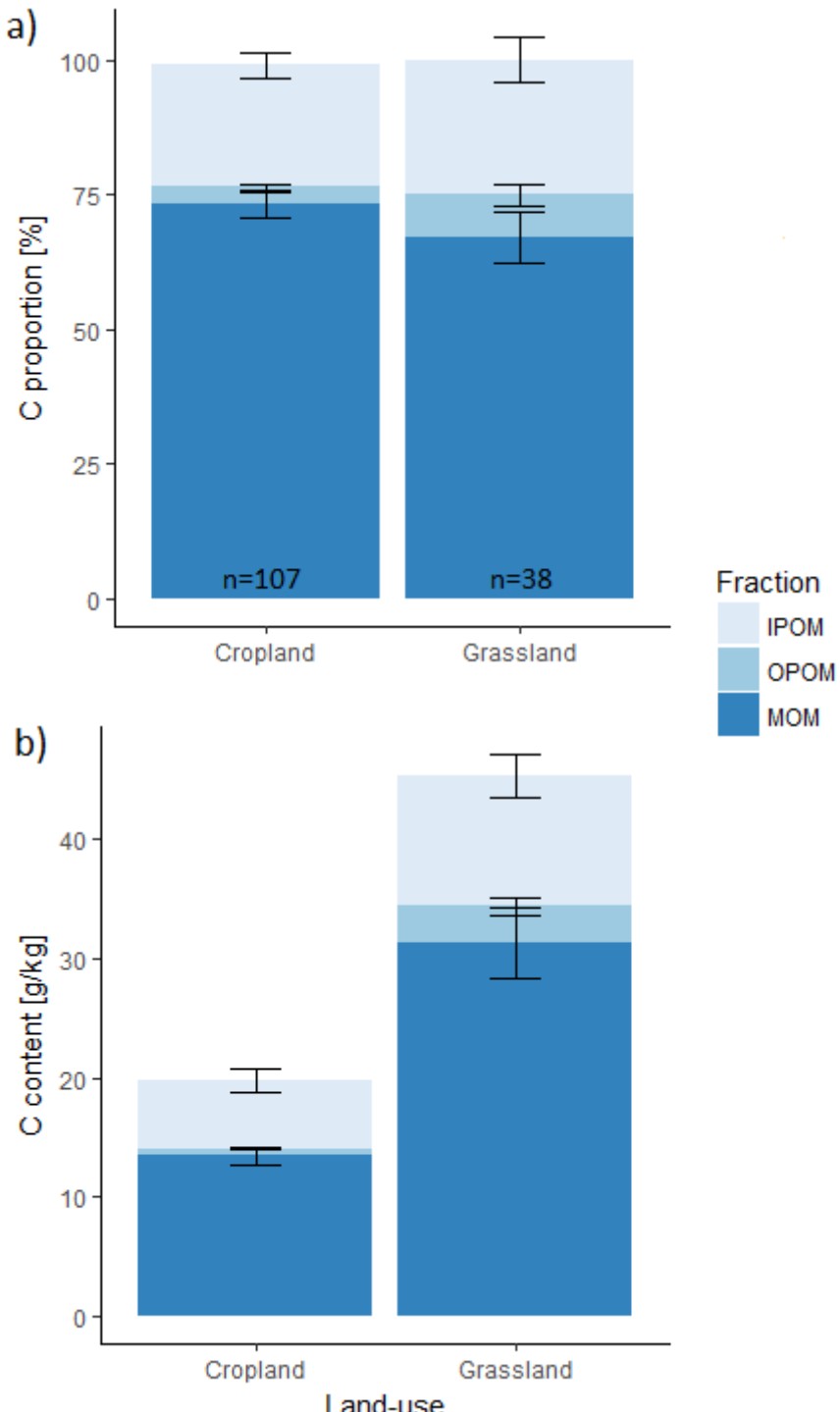

Fig. 1: a) Proportion (%) and b) absolute content (g kg⁻¹) of soil organic carbon (SOC) in the intra-aggregate particulate organic matter (iPOM), occluded particulate organic matter (oPOM) and mineral-associated organic matter (MOM) fraction in soils under cropland and grassland for the 145 calibration sites that were fractionated. Error bars denote standard error of the mean.

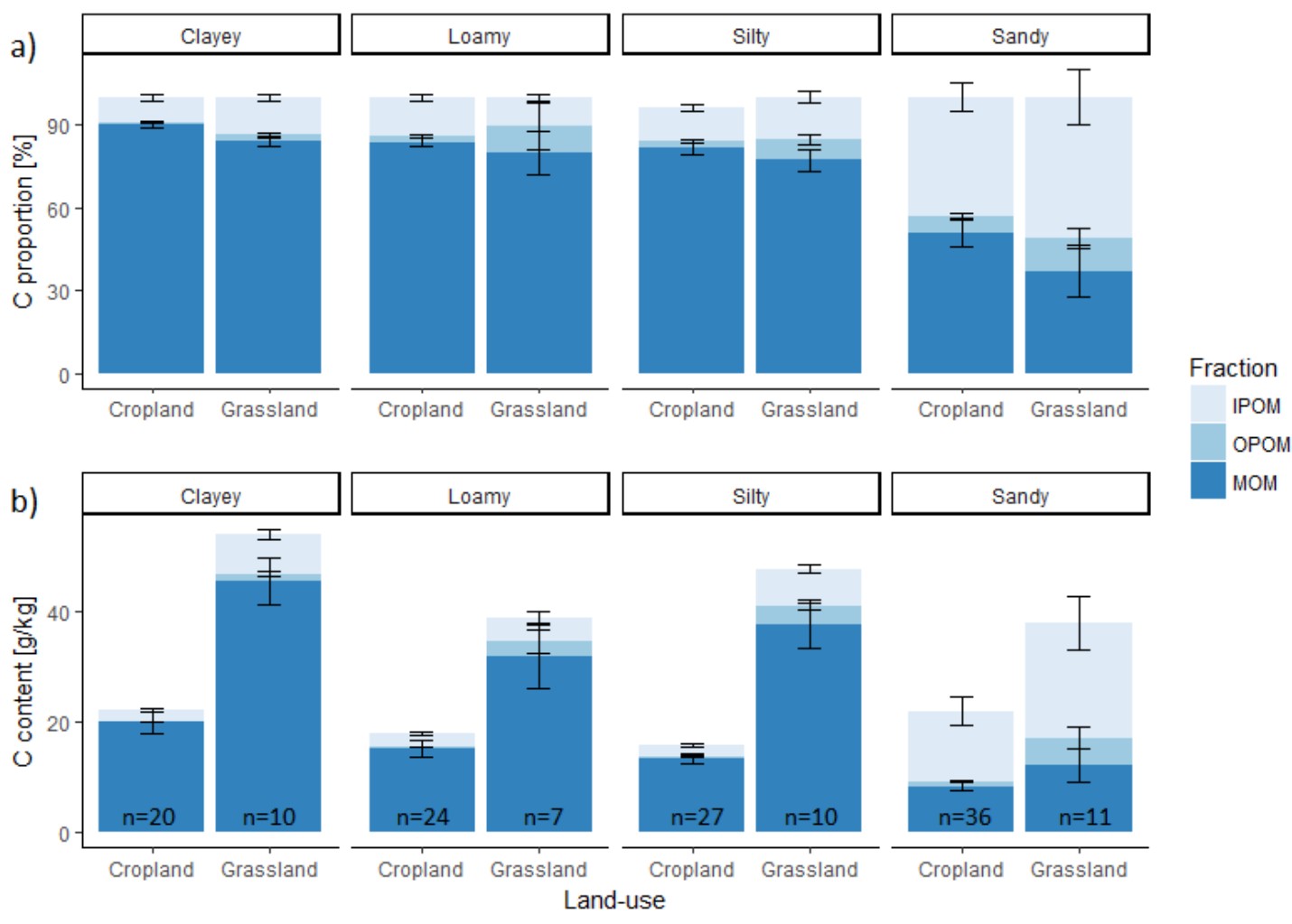

Fig. 2: a) Proportion (%) and b) absolute content (g kg$^{-1}$) of soil organic carbon (SOC) in the intra-aggregate particulate organic matter (iPOM), occluded particulate organic matter (oPOM) and mineral-associated organic matter (MOM) fraction in different soil texture classes for the 145 calibration sites that were fractionated. Error bars denote the standard error of the mean.

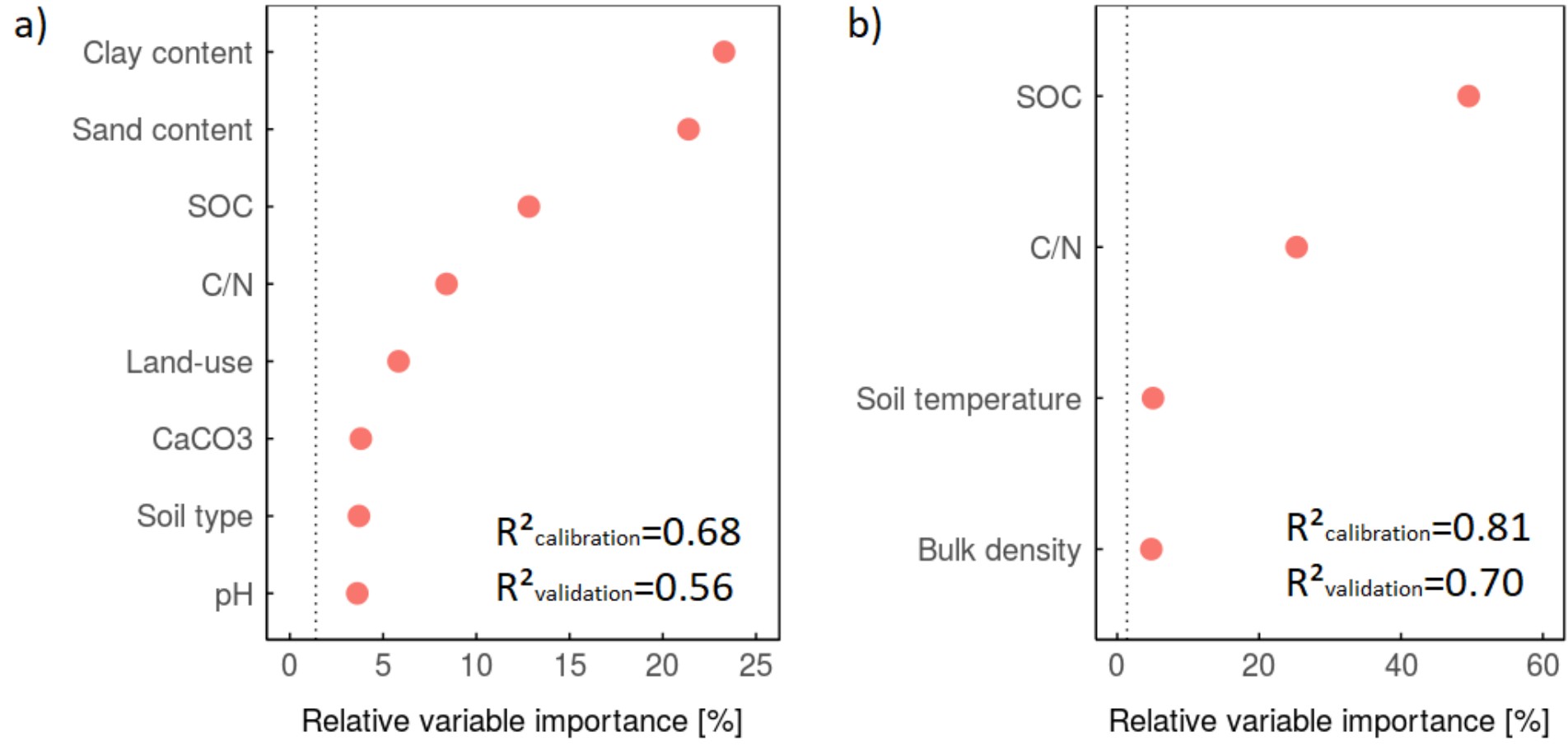

*Fig. 3: Mean relative variable importance according to conditional inference forest (cforest) algorithm for predicted proportion of soil organic carbon (SOC) in the light fraction. The vertical line indicates the threshold value of relative variable importance above which a variable was regarded as important. a) Variable importance for all soils that are not black sands and b) variable importance for only black sands.*

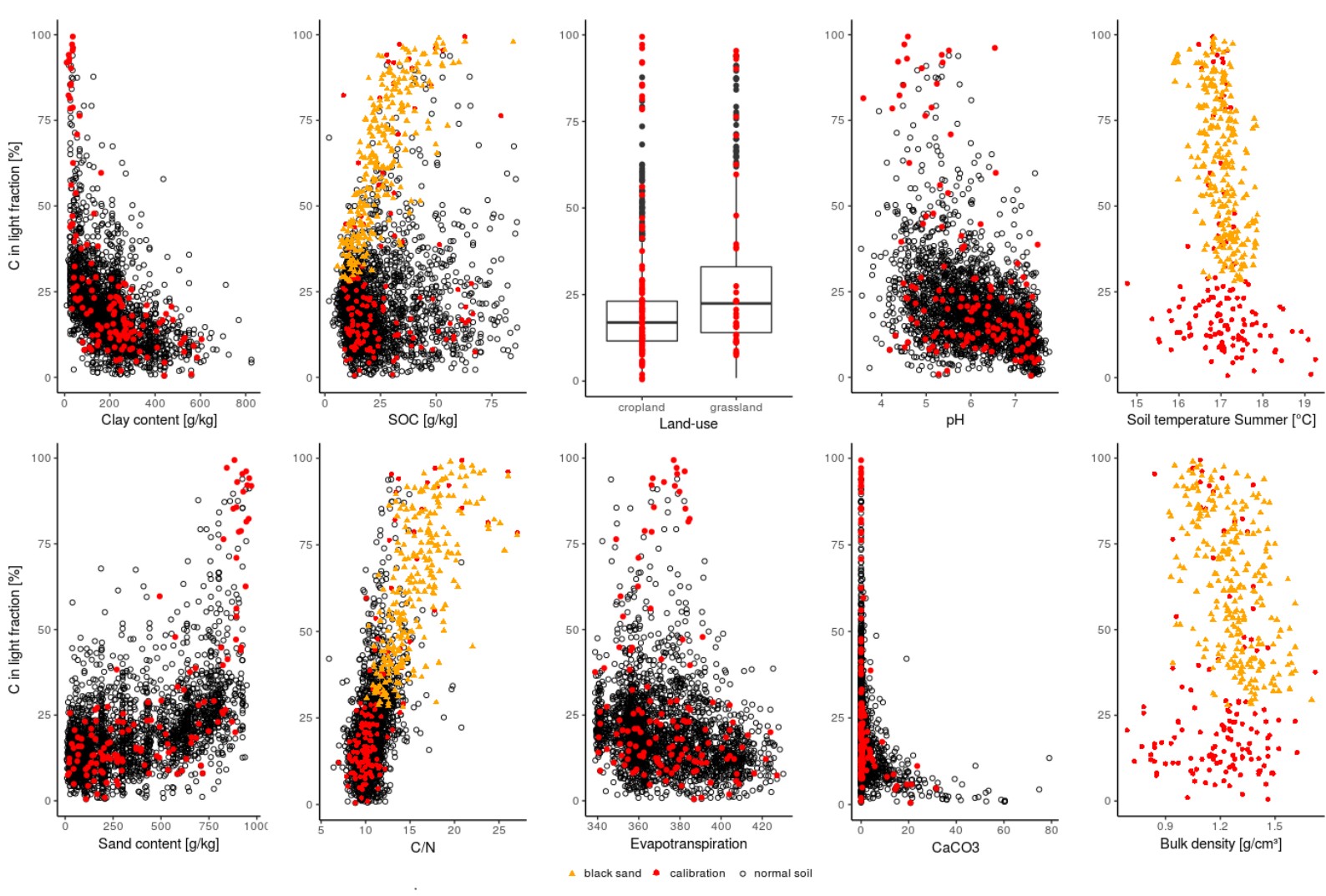

Fig. 4: Relationship between soil organic carbon (SOC) proportion in the light fraction and influential variables. Calibration sites are shown as red dots, normal soils as black dots and black sands as orange triangles.

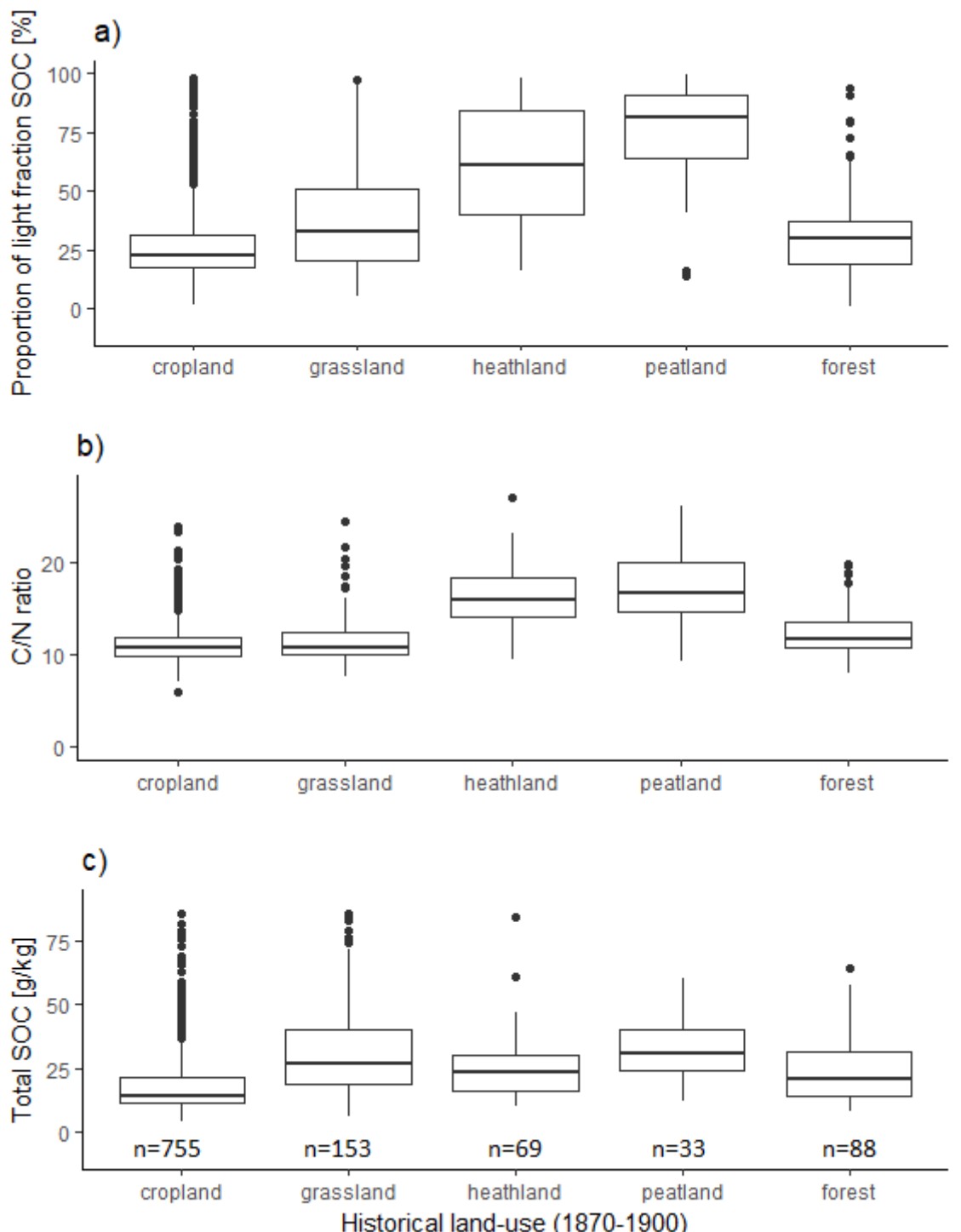

Fig. 5: Relationship between land-use history and a) proportion of light fraction soil organic carbon (SOC), b) carbon/nitrogen (C/N) ratio and c) total SOC content for all sites in the federal states of Lower-Saxony ,Mecklenburg-Western Pomerania, North-Rhine Westphalia, Saxony-Anhalt, Rhineland-Palatinate and Schleswig-Holstein.

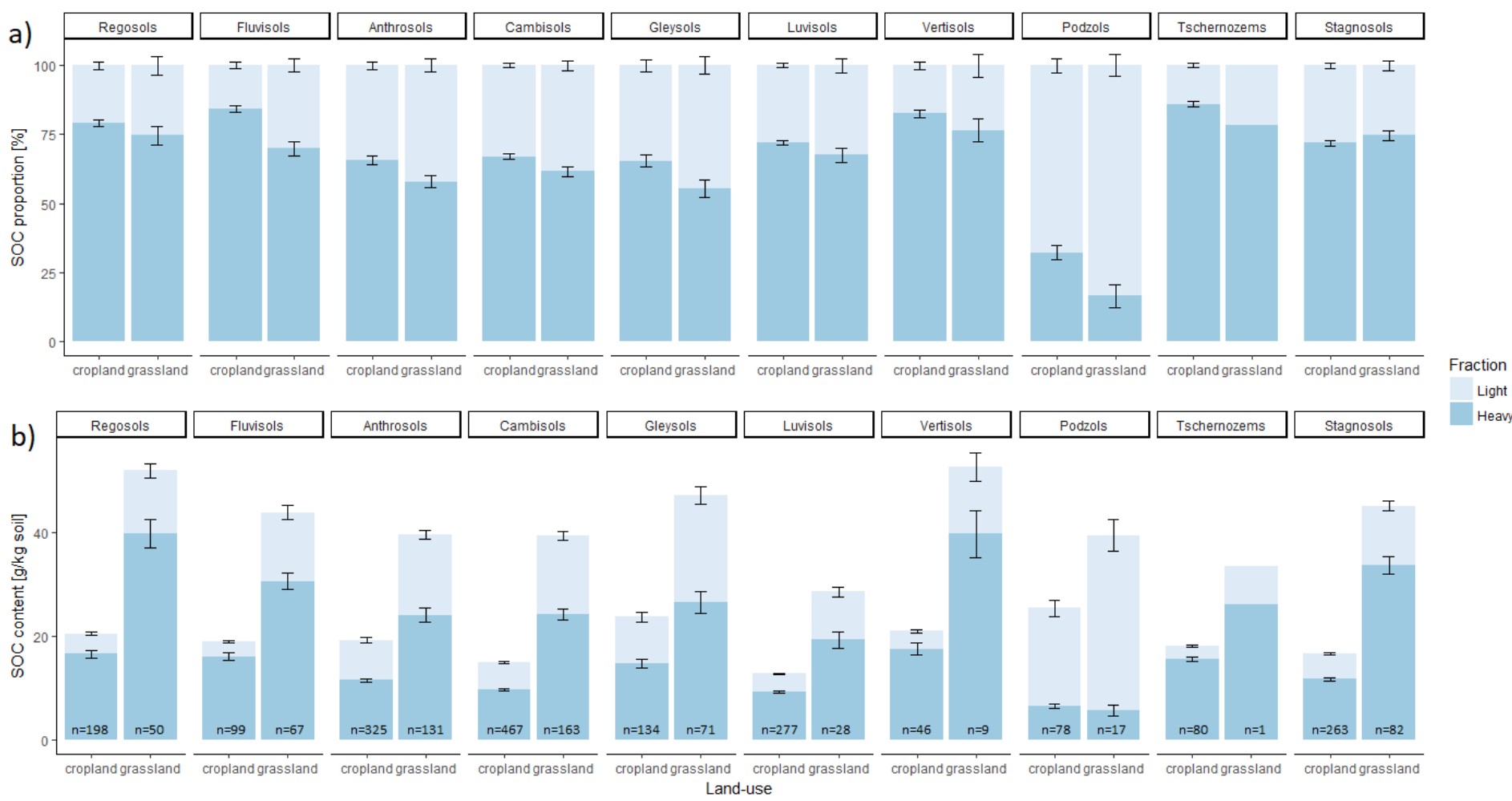

Fig. 6: a) Proportion (%) and b) absolute content (g kg⁻¹) of soil organic carbon (SOC) in the light and heavy fractions in different soil types in the 'normal' soils (non-black sands) dataset. Error bars denote standard error of the mean.

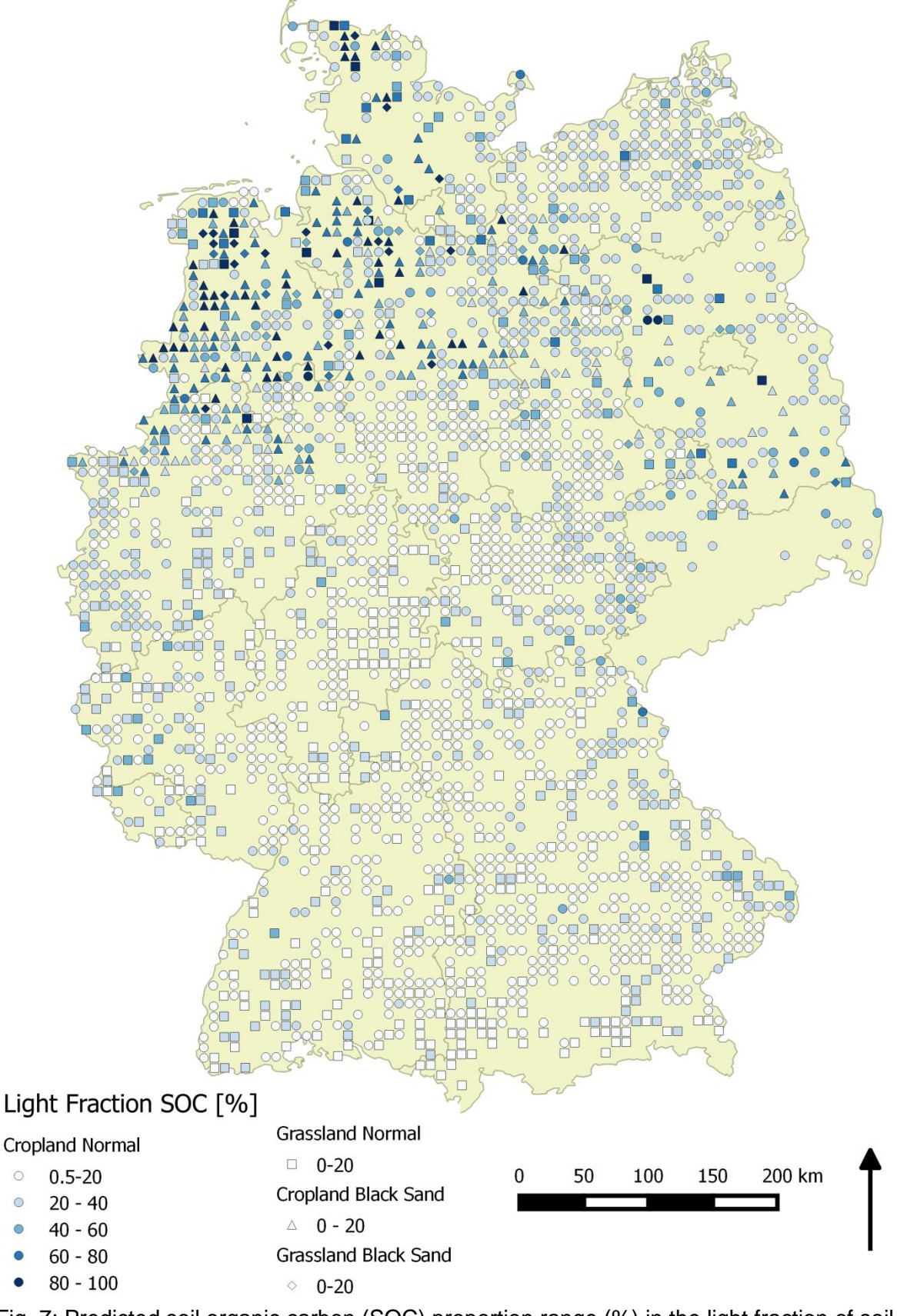

Light Fraction SOC [%]

Cropland Normal
○ 0.5-20
○ 20 - 40
○ 40 - 60
● 60 - 80
● 80 - 100

Grassland Normal
□ 0-20

Cropland Black Sand
△ 0 - 20

Grassland Black Sand
◇ 0-20

0   50   100   150   200 km

Fig. 7: Predicted soil organic carbon (SOC) proportion range (%) in the light fraction of soil at sites in the German Agricultural Soil Inventory.

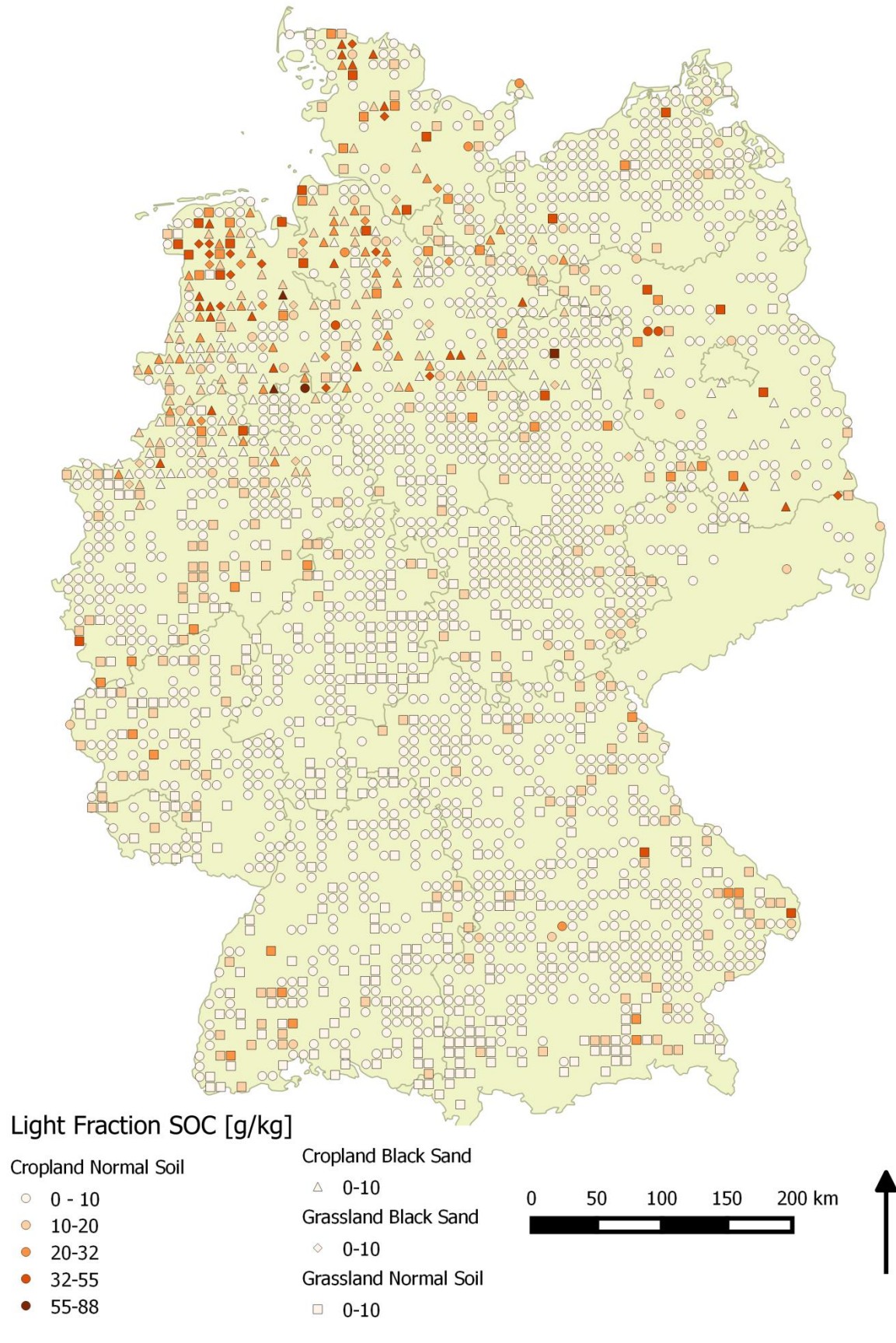

Light Fraction SOC [g/kg]

Cropland Normal Soil
- ○ 0 - 10
- ○ 10-20
- ○ 20-32
- ○ 32-55
- ● 55-88

Cropland Black Sand
- △ 0-10

Grassland Black Sand
- ◇ 0-10

Grassland Normal Soil
- ▢ 0-10

0   50   100   150   200 km

Fig. 8: Predicted absolute soil organic carbon (SOC) content range (g kg$^{-1}$) in the light fraction at sites in the German Agricultural Soil Inventory.

## Supplementary Material

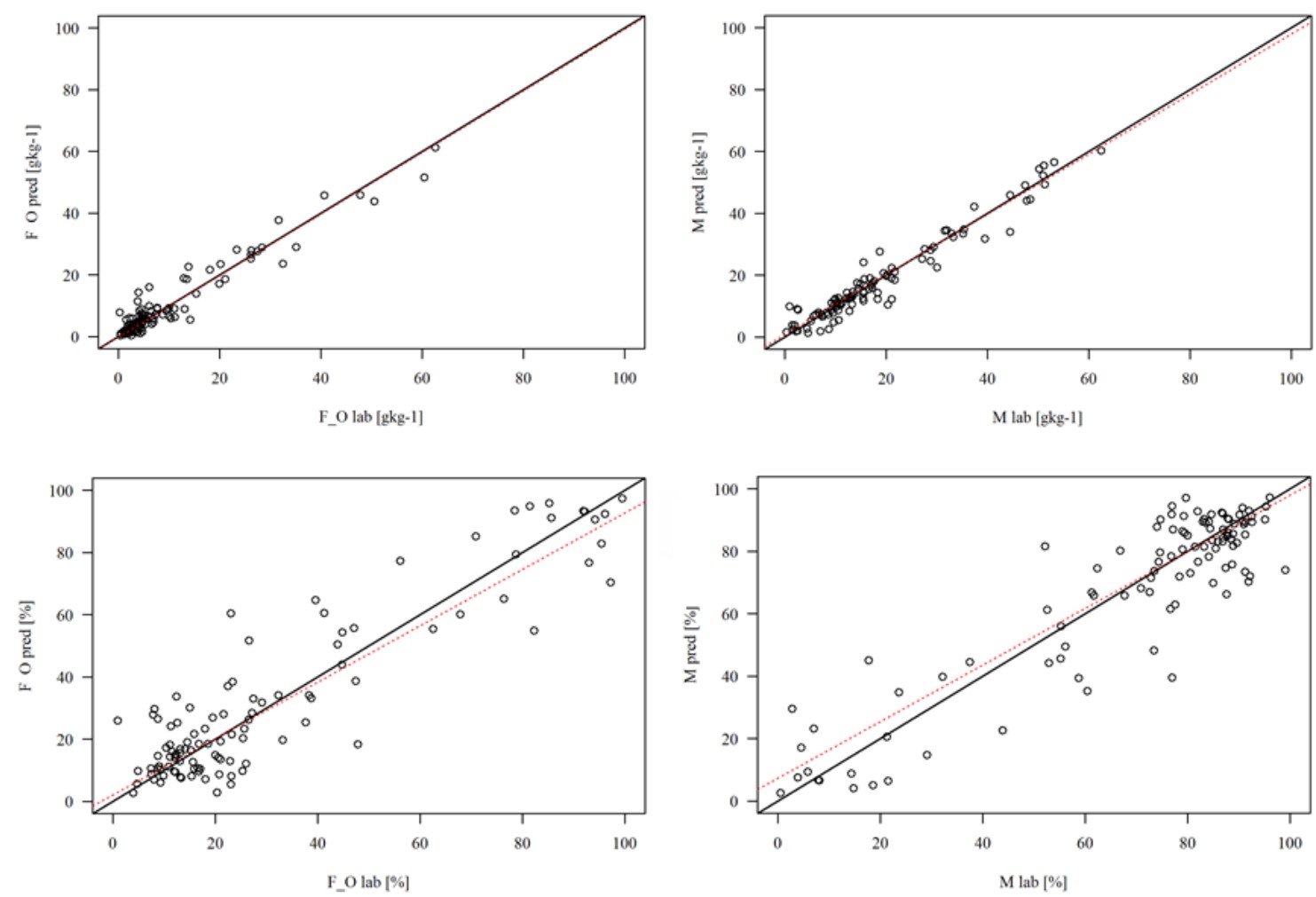

*Figure S1: Measured (lab) versus predicted (pred) values for absolute content (g/kg) and proportion (%) of soil organic carbon (SOC) in fractions. M denotes the MOM fraction, whereas FO denotes the light fraction (iPOM and oPOM)*

*S2: Table of all predictors used for the cforest prediction*

| Driver | Variable type (no. of categories) | Explanation |
|---|---|---|
| Preuss_Nutzung1 | categorical (6) | Historical land-use (1870-1900) |
| K1950_Nutzung1 | categorical (6) | Historical land-use (1950) |
| K1970_Nutzung1 | categorical (6) | Historical land-use (1970) |
| K1990_Nutzung1 | categorical (6) | Historical land-use (1990) |
| BT_Bewirtet | integer | Length of time that the present farmer has farmed this field |
| BT_OekoWirt | categorical (2) | Conventional or organic farming |
| BP_Kalkung | categorical (2) | Does the soil receive lime? |
| BP_Stickstoff | categorical (2) | Does the soil receive mineral N fertiliser? |
| Landnutzung_aktuell | categorical (2) | Current land-use |
| EC_H2O | numeric | Soil electrical conductivity |
| pH_CaCl2 | numeric | Soil pH measured in $CaCl_2$ |
| TOC | numeric | Soil SOC content |
| C_N_Verhaeltnis | numeric | Soil C/N ratio |
| CaCO3 | numeric | Soil carbonate content |
| TRD_FB | numeric | Soil bulk density |
| Wassergehalt | numeric | Soil water content |
| Neigung | integer | Slope of sample point |
| Exposition | categorical (8) | Exposition of sample point |
| Woelbung | categorical (9) | Curvature of sample point |
| Microrelief | categorical (7) | Microrelief of sample point |
| LageImRelief | categorical (9) | Relief position of sample point |
| BodenAbtrag | categorical (3) | Has there been soil removal? |
| AnthropoVeraen | categorical (5) | Have anthropogenic disturbances taken place? |
| Bodenfeuchte | categorical (5) | Soil moisture at sampling |
| Gefuegeform1 | categorical (11) | Soil aggregation1: Spatial distribution of aggregates |
| Gefuegeform2 | categorical (13) | Soil aggregation2: Type of aggregates |
| Risse | categorical (8) | Width of cracks in soil horizon |
| RoehrenArt | categorical (5) | Type of tubes in soil horizon |
| RoehrenBelebt | categorical (7) | Are tubes in soil horizon occupied? |
| RoehrenFlaeche | categorical (7) | Surface proportion of tubes in soil horizon |
| Feinwurzel | numeric | Mass proportion of fine roots |
| GrobWurzel | numeric | Mass proportion of thick roots |
| SumSkelett | numeric | Estimated stone content in soil horizon |
| Substanziell1 | categorical (2) | Substantial soil inhomogeneities |
| Strukturell1 | categorical (4) | Structural soil inhomogeneities |
| Stratigraphie | categorical (18) | Stratigraphy |
| GrundwaStufe | categroical (8) | Groundwater class |

| | | |
|---|---|---|
| GrundwaStand | numeric | Groundwater table |
| Moormaechtig | numeric | Peat thickness |
| BodentypKlasse | categorical (14) | Class of soil type |
| chep | numeric | C export through main crop products |
| cnep | numeric | C inputs through byproduct |
| cewr | numeric | C inputs through roots |
| cod | numeric | C inputs through organic fertiliser |
| nhep | numeric | N export through main crop products |
| nnep | numeric | N inputs through byproducts |
| newr | numeric | N inputs through roots |
| nod | numeric | N inputs through organic fertilisers |
| nmin | numeric | N inputs through mineral fertilisers |
| EvapotransPot | numeric | Potential evapotranspiration |
| EvapotransReal | numeric | Real evapotranspiration |
| DroughtIndexMean | numeric | Drought index |
| PrecYearMean | numeric | Mean annual precipitation (30 y mean) |
| TempYearMean | numeric | Mean annual temperature (30 y mean) |
| SoilMoistSummer | numeric | Soil moisture in 5 cm soil depth in summer |
| SoilTempSummer | numeric | Soil temperature in 5 cm depth in summer |
| NDVI_July | numeric | Mean NDVI in July |
| slope_100 | numeric | Slope from digital elevation model with resolution 100m |
| topoidx_100 | numeric | Topographical wetness index from digital elevation model with resolution 100 m |
| BodenAusMatKlasse | categorical (14) | Class of parent material |
| LN | categorical (7) | Reported land-use changes |
| MR | categroical (5) | Meliorative management measures |
| Jahre_wendend | integer | Number of years with full inversion tillage over the past 10 years |
| Jahrenichtwendend | integer | Number of years with reduced tillage over the past 10 years |
| Jahre_Getreide | integer | Number of years with grains in the rotation over the past 10 years |
| Jahre_FeldgrasKlee | integer | Number of years with clover in the rotation in the last 10 years |
| gleicheKultur5Jahre | integer | Where there five or more consecutive years with the same crop grown? |
| Anz_Kulturgruppen | integer | Number of different crops grown in last 10 years |
| Schluff | numeric | Soil silt content |
| Ton | numeric | Soil clay content |
| Sand | numeric | Soil sand content |

Table S3:

Indicators of model performance for soil C fractions particulate organic carbon (POM) and mineral associated organic carbon (MOM) with calibration and independent validation dataset (mean values of 100 iterations with random selection). Table a) is for values in g C kg soil$^{-1}$ and table b) is for the proportion (relative values).

a)

| | Calibration dataset | | | | | | Validation dataset | | | | | |
| | $Q^2$ | RMSECV, g C kg soil$^{-1}$ | $\rho c_c^*$ | Bias, g C kg soil$^{-1}$ | RPD | RPIQ | $R^2$ | RMSEP, g C kg soil$^{-1}$ | $\rho c_v$ | Bias, g C kg soil$^{-1}$ | RPD | RPIQ |
|---|---|---|---|---|---|---|---|---|---|---|---|---|
| POM | 0.83 | 4.92 | 0.91 | 0.34 | 2.5 | 1.8 | 0.82 | 5.38 | 0.89 | 0.44 | 2.5 | 2.0 |
| MOM | 0.87 | 4.92 | 0.93 | -0.34 | 2.9 | 2.9 | 0.85 | 5.38 | 0.91 | -0.44 | 2.7 | 2.6 |

$\rho c^*$ - Lin's concordance correlation coefficient

b)

| | Calibration dataset | | | | | | Validation dataset | | | | | |
| | $Q^2$ | RMSECV, % | $\rho c_c^*$ | Bias, % | RPD | RPIQ | $R^2$ | RMSEP, % | $\rho c_v$ | Bias, % | RPD | RPIQ |
|---|---|---|---|---|---|---|---|---|---|---|---|---|
| POM | 0.78 | 13.15 | 0.88 | 1.07 | 2.09 | 2.56 | 0.73 | 15.04 | 0.84 | 1.6 | 1.9 | 2.4 |
| MOM | 0.78 | 13.15 | 0.88 | -1.07 | 2.00 | 2.48 | 0.72 | 15.04 | 0.83 | -1.6 | 2.0 | 2.3 |

$\rho c^*$ - Lin's concordance correlation coefficient