# Peer review of "Hot regions of labile and stable soil organic carbon in Germany - Spatial"

_SOIL, 2017_

## Short Comment (SC1) · 15 Jan 2018

Comment on "Hot regions of labile and stable soil organic carbon in Germany - Spatial variability and driving factors" by Vos et al. (SOIL Discuss.)

Lauric Cécillon1,2, Bernard Barthès3, Suzanne Lutfalla2, Laure Soucémarianadin2, Pierre Barré2

1 Université Grenoble Alpes, Irstea, UR LESSEM, 2 rue de la Papeterie, 38402 St-Martin-d'Hères, France 2 Laboratoire de Géologie, PSL Research University, CNRS-ENS UMR 8538, Ecole Normale Supérieure, 24 rue Lhomond, 75231 Paris CEDEX 5, France 3 IRD, UMR Eco&Sols, Montpellier SupAgro, 2 place Viala, 34060 Montpellier Cedex 2, France

In their interesting and stimulating paper, Vos and colleagues aimed at studying the distribution of labile and stable soil organic carbon (SOC) fractions in a large set of agricultural (cropland and grassland) topsoils (0–10 cm depth) from the German Agricultural Soil Inventory (2900 sites).

They define the labile SOC fraction as low density (d < 1.8) particulate organic matter (POM) carbon, while the stable SOC fraction is defined as high density (d > 1.8) mineral-associated organic matter (MOM) carbon. Both labile and stable SOC fractions were isolated by a soil organic matter density fractionation scheme on a subset of 145 samples.

They then calibrated using this subset of 145 samples a multivariate regression model relating the absolute content (g/kg) and proportion (%) of SOC in the POM and in the MOM fractions to soil absorbance in the near-infrared spectral domain (predictor variables; 1300–3300 nm). They used a leave-one-out cross-validation to validate their model, and briefly reported the cross-validated predictive performance of the model in the Material & methods section of the paper and in the Supplementary Figure S1. The authors mentioned that more details of the multivariate regression model based on near-infrared spectroscopy can be seen in a submitted paper (Jaconi et al., submitted) with no reference to the journal and/or submission tracking number of this manuscript.

The authors then used their multivariate regression model based on near-infrared spectroscopy to predict the absolute content (g/kg) and proportion (%) of SOC in the POM and in the MOM fractions of the remaining topsoils of the German Agricultural Soil Inventory (n = 2755).

Finally, Vos and colleagues used a random forest algorithm to investigate the relative importance of 75 potential drivers of differences in carbon proportions in the labile and stable pools using both the calibration set (n = 145) and whole dataset (n = 2900).

We have a concern regarding the use of the cross-validated regression model based on near-infrared spectroscopy to predict the size of SOC labile and stable pools in Interactive comment

"new" samples of the German Agricultural Soil Inventory.

We regret the use a regression model that has not been published yet, impeding us from a clear understanding of the actual predictive performance of the model on "new" topsoil samples. Here, the details provided by the authors regarding the predictive performance of the multivariate regression model (see Material & methods section 2.4 at lines 189–194 and Supplementary Figure S1) do not demonstrate its ability to accurately predict the absolute content (g/kg) and proportion (%) of SOC in the POM and in the MOM fractions of the 2755 "new" samples.

Specifically, the authors have only assessed the predictive performance of their model using a leave-one-out cross-validation. Leave-one-out cross-validation is not the optimal method to validate a partial least-squares (PLS) regression model when 145 samples with reference measurements are available. It may be recommended for smaller datasets when a proper validation procedure (see below) cannot be done.

An acceptable procedure for validating this PLS regression model would be adding an independent validation step to the current validation scheme:

i/ first run a leave-one-out or k-fold cross-validation on a subset of ca. 110 samples with reference measurements, that would provide a  $Q^2$  (= coefficient of determination of the model in cross-validation, not a  $R^2$ ), and a first assessment of the mean error of prediction of the PLS regression model in cross-validation (RMSECV).

ii/ use this cross-validated PLS model to predict the values of the absolute content (g/kg) and proportion (%) of SOC in the POM and in the MOM fractions of the ca. 35 independent samples with reference measurements not used for cross-validation (and independent from the ca. 110 samples used for cross-validation). The coefficient of determination (actual coefficient of determination of the model in validation, R2) and mean error of prediction of the PLS regression model in validation (RMSEP) would provide acceptable criteria for the reliable (independent) assessment of the actual predictive performance of the model for prediction on "new" topsoil samples.
iii/ if the R2 and RMSEP (or RPD) of the PLS regression model obtained on the 35 independent validation samples were judged acceptable, then the model may be used to predict the values of the absolute content (g/kg) and proportion (%) of SOC in the POM and in the MOM fractions of the 2755 remaining topsoils of the German Agricultural Soil Inventory.

We therefore argue that the PLS regression model based on near-infrared spectroscopy presented by the authors cannot be used in its current form to predict labile and stable SOC fractions on "new" topsoil samples of the German Agricultural Soil Inventory.

At this stage (i.e. unreliable assessment of the predictive performance of the PLS regression model), the authors can only use the reference data (n = 145) of the absolute content (g/kg) and proportion (%) of SOC in the POM and in the MOM fractions to investigate the potential drivers of the distribution of SOC kinetic pools on this limited dataset. This would already be a significant piece of work.

Furthermore, Vos and colleagues used the particulate organic matter (POM) fraction to represent the labile SOC kinetic pool. However, the POM fraction could contain substantial (and variable) amounts of pyrogenic carbon with residence time in soils higher than the mean residence time of total SOC. This limitation of the SOC density fractionation scheme should be mentioned and discussed in the text, as it is not possible to guaranty that the POM fraction truly represents the actual labile SOC pool for all investigated samples.

We are looking forward to seeing an improved version of this work in SOIL.

---

## Referee Comment (RC1) · Anonymous Referee #1 · 18 Jan 2018

This is an interesting study on the spatial distribution of SOC fractions (POM and fine fraction) in agricultural soils of Germany. The authors fractionated 145 selected sites und used this dataset to calibrate NIRS predictions for more than 2500 sites. Conditional inference tree modelling was then used to identify the driving factors of fractions based on 75 factors related to soil properties, management, land use (history) and environmental aspects. This well-written paper is the first that provides insights into the driving factors of SOC fractions at the regional scale. The methodology including NIRS predictions for fractions and cforest to elucidate driving factors is novel and can also be used in future similar studies in other regions. Overall, this paper is a highly valuable contribution to soil organic matter research at the regional scale and provides important information for improved SOC management. Nevertheless, there are sev-

eral points which have to be addressed before the manuscript can be accepted for publication:

L62: There is ample evidence that no-till does not lead to net increase of SOC compared to conventional tillage as indicated here, but only to a change of the depth distribution of SOC

Section 2.3: The fractionation approach is not really clear: to separate the fPOM, normally SPT is used as done in this study, but without any dispersion (as indicated by "free"). Here, ultrasonic dispersion at 65 J/mL was applied that probably destroyed macroaggregates, so the extracted POM is rather fPOM+oPOM (derived from macroaggregates). Of course you can do that, but this fraction should not be called fPOM. Furthermore, 450 J/mL was used to destroy "aggregates" (I guess microaggregates), please explain why this energy level was used (reference). I further miss information on recovery rates of the fractionation and further basic data such as fraction mass and C content in order to evaluate the approach. Particularly the measured C content of the POM is important to evaluate the fractionation approach.

L182+L194: More information is needed in this regard, Jaconi et al. is not accessible (see also the comment of Lauric Cécillon). Please include relevant information also in this paper, even if Jaconi et al. is published during revision.

L188-198: I would rather see that as results

L197-198: NIRS is certainly a promising way to predict fractions, but of course this approach is specific to the fractionation. As there are numerous other fractionation approaches (probably even better ones to derive "active" and "passive" SOC), this study should avoid giving the impression that the presented approach is the only way to estimate active and passive SOC at the regional scale.

L203-205: More information is needed on the calculation of C and N inputs as well as on the regional yield estimates.

L229: In order to avoid interaction effects between the variables, one could perform PCAs prior to the analysis and reduce the number of predictors to independent ones (e.g. dependent climate variables MAT, MAP and elevation can be reduced to one factor climate). For example, CaCO3 was identified as important, but this is probably only due to a correlation with texture (clay is the most important factor).

L316: remove "and"

Section 4.4: In principal, I agree that the fractionation approach based on a separation of POM from MOM is suitable to derive "labile" and "stable" carbon, as POM is the major constituent of "active" carbon (assuming that the contribution of pyrogenic carbon is negligible, which is the case in most regions of Germany). However, the authors could mention that there are other ways to derive labile and stable SOC.

---

## Referee Comment (RC2) · A. P. Smith (Referee) · 16 Feb 2018

Vos et al. present a region-wide investigation of SOC content among fractions and analyses into potential drivers of SOC content. They used a novel approach by using NIRS to predict C fractions and were able to describe POM and MOM contents and drivers across multiple sites in Germany. This work warrants publication in SOIL, but is not publishable in its current state. There are many points to be addressed and I recommend some substantial rewriting and reorganization of the paper. Please see below comments as well as comments in text (uploaded as an attachment).

Introduction

Overall, I think that the introduction needs some restructuring and needs more "meat"

[Figure]

to it. Many statements are vague, blanket statement and don't provide much insight or examples (e.g. "The effects of land use and management are not the same for all soil organic matter compounds..." How? Why? Give me more details). I think that the manuscript would benefit from a closer look at the flow and organization of the introduction. I suggest taking a close look at each paragraph; map out the main point, make sure this main point is reflected in the topic sentence, and verify that the preceding and following paragraphs fit/flow. There are a few paragraphs that just don't fit (seem out of place) and it detracts from the main points of the introduction (which is essentially to build up to, i.e. provide background and rationale, the objectives and hypotheses of the study). As such, please align the introduction specific to the goals and objectives of the study.

I strongly encourage the authors to reframe the objectives of the study as hypotheses in lieu of the somewhat vague research questions that are currently reported in the introduction. What do the authors expect the distribution of POM vs. MOM to be across Germany (and why)? Which factors (land-use, climate, soil type, clay content, etc.) do the authors expect to be more important in driving these distributional patterns? And the final question "can regions of high vulnerability..." needs to be clarified. First, I don't know how you define "vulnerable" and second, I am unaware how you plan on verifying that your predictive approach (i.e. machine learning)

Many of the statements or research addressed here are specific to European agro-ecosystems and yet the authors often make broad statements about land use and management effects on SOC as fact. However, land use and management effects on SOC differ greatly depending on cropping system, location (climate, topography, parent material, etc.) and there is often an equal amount of work that supports different results than what you present in this paper. As such, please be more specific and make sure to constrain postulations with "in temperate cropping systems..." or something to that example. I would be satisfied with a sentence early on stating that you are limiting the state of art (or body of knowledge) to your specific system (i.e. western European

cropping systems).

As mentioned earlier, many sentences are vague. Please try to be more specific and detailed when building up the background and rationale in the introduction. There is more "telling" than "showing". Please see the attached line-by-line review.

Methods

Overall, I suggest reorganizing the methods section to be more aligned with your objectives. This is especially true when it comes to the use of calibration versus all samples. Sections often jump from calibration to all and it makes it a bit confusing. There also needs to be more technical details into how soils were collected and processed (e.g. collected with a corer, composite samples, one sample per depth, homogenized, dried, etc. ?). Replication need to be explicitly stated (how many samples did you use for each classification combination – i.e. land use, or depth, etc.). Including a supplemental table that lists all the samples/sites or something may help clear this up. There are also several areas where the methods need to be more explicitly stated and many instances were citations are needed. Please see attachment for line by line comments.

Calibration samples versus all: The experimental design (use of calibration sites versus all sites) needs to be clearer. It was confusing with the way the methods section was organized for the reader to understand why/what/how calibration samples were used as compared to all sites. Perhaps have a separate calibration section in the methods where all of this is addressed would be clearer.

A major issue I have with the methods is combining the oPOM and fPOM fractions together as a "light fraction." As much as I hate to ask authors to redo their analyses, I think that the best way to deal with the oPOM is to either ignore it or analyze it separately.

Results

Please review my comments in the attachment and address them. Most importantly, I

do not agree with using total SOC to explain fraction SOC. Of course, C would explain C. Total SOC is NOT a driver – it is a response variable for this study.

You are also missing any reference to Fig. 6 and Fig. 8 in the results! If you don't use them – don't put them in the manuscript (or put them in supplemental).

Discussion

I would almost reorganize the discussion to be more explicitly aligned with the study objectives – first discuss the how SOC is distributed among fractions at a national scale, then discuss which drivers are relevant and finally end with whether or not you can predict "vulnerable" (but please define) areas using your approach. Section 4.1 is entirely too brief, especially since it supposedly addresses your first objective. Again – don't just tell me what other results support or do not support your results, show me!

You have a great discussion on the "black sands" section. I would love to see that reflected in the entire discussion section. Some of the details I was looking for in section 4.1 are included in 4.2. I think it would be good to combine section 4.1 and 4.2 (and address your first objective) and discuss black sands in the context of objective 1.

In section 4.4, it would be great to discuss why/why not you think your approach worked to identify vulnerable areas. It is one of your objectives and you do not directly discuss it in the discussion. It needs to be addressed. I think concluding section 4.4 with a paragraph answering "Can regions of high vulnerability to carbon losses be identified by this predictive approach?" is warranted.

Conclusion

See notes regarding final sentence.

I believe that with a few revisions (as per my and other reviewers' suggestions) this manuscript is publishable and I look forward to the revisions!

A. Peyton Smith Soil Biogeochemistry & Microbial Ecology Pacific Northwest National

Laboratory

Please also note the supplement to this comment:
https://www.soil-discuss.net/soil-2017-30/soil-2017-30-SC2-supplement.zip

---

## Short Comment (SC3) · 16 Feb 2018

I thank the authors for their paper and I hope that my discussion helps. My comments here relate primarily to the lack of clarity in the description of the methods used for the spectroscopic modelling, and to missing quantification of robustness and uncertainty in the spectroscopic model predictions of the carbon fractions. I believe these to be crucially important because their further analyses and interpretation of the variability and driving factors relies heavily on the spectroscopic model predictions.

First, the description of the spectroscopic modelling is inadequate and I encourage the authors to improve it. I think that the specifics of the spectroscopic modelling, apparently described in Jaconi et al., need to be included in this manuscript, particularly

because the Jaconi et al. manuscript isn't yet published. But, even if the Jaconi et al paper were published, I think that at the very least, readers will need a clear summary of their methods and findings–not simply a report of their assessment statistics.

Second, the authors do not convincingly show that the spectroscopic models were sufficiently robust for predicting the 'unknowns', which I presume were the '. . .>2500 sites with mineral soil all over Germany' (mentioned only in the Introduction, line 106). Additional validation of the models with an independent test set will help, however, I would also encourage the authors to implement either a repeated cross validation, or to bootstrap the models to quantify their robustness and the uncertainty of their predictions (see for instance Viscarra Rossel, 2007). To this end, the authors might find it useful to read Viscarra Rossel & Hicks (2015). There, we proposed an approach for modelling the carbon fractions of a large continental scale dataset, reporting the robustness of the models, the (propagated) uncertainties of the predictions, and relating the spectroscopy to the chemistry of soil organic C.

Quantifying uncertainty is particularly important when predicting 'unknown' samples. Without quantified uncertainty, the predictions will definitely be less valuable. This is particularly relevant for this study because the predictions are being used in subsequent analysis to potentially gain new understanding.

Finally, I would like to suggest some minor corrections:

- In lines 182–183, the Jaconi et al reference is cited as 'in prep' while in line 194 it is cited as 'submitted'

- The mention of the '. . .>2500 sites with mineral soil all over Germany.', in the Introduction, line 106, is inadequate. This should be described and made clear in the Methods section–possibly in section 2.4 after a (better) description of the spectroscopic modelling.

- In lines 185–187: '. . . In addition, residual prediction deviation (RPD) was calculated,

using the classification system devised by Viscarra Rossel et al. (2006)....' – I am quite sure that Viscarra Rossel et al. (2006) did not devise a classification for the RPD. Williams (1987) originally devised the RPD for assessing spectroscopic calibrations of agricultural and food products. Later, Chang et al. (2001) suggested an arbitrary classification specifically for soil. It is very likely that Viscarra Rossel et al. (2006) simply used that classification, but I could not confirm one way or the other because the Viscarra Rossel et al. (2006) reference is not listed in the references.

- In terms of the RPD, Bellon-Maurel et al. (2010) suggested that the RPD should only be used if the data is normally distributed, otherwise, they propose the use of the RPIQ (Bellon-Maurel et al., 2010).

- Following from that, in our spectroscopic modelling of soil carbon and fractions (Viscarra Rossel & Hicks, 2015), we found that their statistical distributions were often not normal and required logarithmic transformations. For this reason, it would be useful for the authors to report the distributions of the carbon and fractions data–but also because the PLSR algorithm assumes normally distributed data.

I hope that this helps.

Raphael VISCARRA ROSSEL

References cited:

Bellon-Maurel, V., Fernandez-Ahumada, E., Palagos, P., Roger, J-M., McBratney, A.B., 2010. Critical review of chemometric indicators commonly used for assessing the quality of the prediction of soil attributes by NIR spectroscopy. TrAC Trends in Analytical Chemistry 29, 1073-1081.

Chang, C.-W., Laird, D.A., Mausbach, M.J., Hurburgh C.R., 2001. Near-infrared reflectance spectroscopy - Principal components regression analyses of soil properties. Soil Science Society of America Journal 65, 480-490.

Viscarra Rossel, RA. 2007. Robust modelling of soil diffuse reflectance spectra by

bagging-partial least squares regression. Journal of Near Infrared Spectroscopy 15: 39–47.

Viscarra Rossel RA, Hicks WS. 2015. Soil organic carbon and its fractions estimated by visible–near infrared transfer functions. European Journal of Soil Science, 66(3): 438–450.

Williams, P.C. 1987. Variables affecting near-infrared reflectance spectroscopic analysis. In: Near- Infrared Technology in the Agricultural and Food Industries (eds P. Williams & K. Norris), pp. 143– 167. American Association of Cereal Chemists Inc., Saint Paul, MN.

---

## Author Response (AR1)

**1) Comments from referees/public and author's response**

**A) Short comment 1**

**Dear Lauric Cécillon and colleagues,**

**Thank you very much for your comment on our discussion paper. We appreciate that you discussed**
**the paper draft thoroughly and found some points that need more clarification to be**
**understandable. Please find our answers to your comments below.**

*We have a concern regarding the use of the cross-validated regression model based on near-infrared*
*spectroscopy to predict the size of SOC labile and stable pools in "new" samples of the German*
*Agricultural Soil Inventory. We regret the use a regression model that has not been published yet,*
*impeding us from a clear understanding of the actual predictive performance of the model on "new"*
*topsoil samples. Here, the details provided by the authors regarding the predictive performance of the*
*multivariate regression model (see Material & methods section 2.4 at lines 189–194 and*
*Supplementary Figure S1) do not demonstrate its ability to accurately predict the absolute content*
*(g/kg) and proportion (%) of SOC in the POM and in the MOM fractions of the 2755 "new" samples.*

**Answer: The paper describing the regression model (Jaconi et al.) has been submitted to the**
**European Journal of Soil Science. We also regret that it has not been published yet. In this paper,**
**the model is described in detail, testing the algorithm on different datasets. In the paper the model**
**is also validated using an independent validation dataset (consisting of one third of the total**
**samples), which has not been part of the model calibration (two thirds of total samples). We see**
**that it would be helpful to provide the validation results with the paper discussed here, as they are**
**not published yet with the other paper. In the revised version we will append the following table**
**with the supplement materials:**

Table S3: Indicators of model performance for soil C fractions particulate organic carbon (POM) and
mineral associated organic carbon (MOM) with calibration and independent validation dataset (mean
values of 100 iterations with random selection). Table a) is for values in g C kg soil$^{-1}$ and table b) is
for the proportion (relative values).

a)

| | $Q^2$ | RMSECV, g C kg soil$^{-1}$ | $\rho c_c^*$ | Bias, g C kg soil$^{-1}$ | RPD | RPIQ | $R^2$ | RMSEP, g C kg soil$^{-1}$ | $\rho c_v$ | Bias, g C kg soil$^{-1}$ | RPD | RPIQ |
|---|---|---|---|---|---|---|---|---|---|---|---|---|
| | | **Calibration dataset** | | | | | | **Validation dataset** | | | | |
| POM | 0.83 | 4.92 | 0.91 | 0.34 | 2.5 | 1.8 | 0.82 | 5.38 | 0.89 | 0.44 | 2.5 | 2.0 |
| MOM | 0.87 | 4.92 | 0.93 | -0.34 | 2.9 | 2.9 | 0.85 | 5.38 | 0.91 | -0.44 | 2.7 | 2.6 |

$\rho c^*$ - Lin's concordance correlation coefficient
b)

| | $Q^2$ | RMSECV, % | $\rho c_c^*$ | Bias, % | RPD | RPIQ | $R^2$ | RMSEP, % | $\rho c_v$ | Bias, % | RPD | RPIQ |
|---|---|---|---|---|---|---|---|---|---|---|---|---|
| | | **Calibration dataset** | | | | | | **Validation dataset** | | | | |
| POM | 0.78 | 13.15 | 0.88 | 1.07 | 2.09 | 2.56 | 0.73 | 15.04 | 0.84 | 1.6 | 1.9 | 2.4 |
| MOM | 0.78 | 13.15 | 0.88 | -1.07 | 2.00 | 2.48 | 0.72 | 15.04 | 0.83 | -1.6 | 2.0 | 2.3 |

$\rho c^*$ - Lin's concordance correlation coefficient

*Specifically, the authors have only assessed the predictive performance of their model using a leave-*
*one-out cross-validation. Leave-one-out cross-validation is not the optimal method to validate a*
*partial least-squares (PLS) regression model when 145 samples with reference measurements are*
*available. It may be recommended for smaller datasets when a proper validation procedure (see*
*below) cannot be done.*

*An acceptable procedure for validating this PLS regression model would be adding an independent*
*validation step to the current validation scheme: i/ first run a leave-one-out or k-fold cross-validation*
*on a subset of ca. 110 samples with reference measurements, that would provide a $Q2$ (= coefficient*
*of determination of the model in cross-validation, not a $R2$), and a first assessment of the mean error*
*of prediction of the PLS regression model in cross-validation (RMSECV). ii/ use this cross-validated PLS*
*model to predict the values of the absolute content (g/kg) and proportion (%) of SOC in the POM and*
*in the MOM fractions of the ca. 35 independent samples with reference measurements not used for*
*cross-validation (and independent from the ca. 110 samples used for cross-validation). The coefficient*
*of determination (actual coefficient of determination of the model in validation,$R2$ )and mean error of*
*prediction of the PLS regression model in validation (RMSEP) would provide acceptable criteria for the*
*reliable (independent) assessment of the actual predictive performance of the model for prediction on*
*"new" topsoil samples.*

*iii/ if the $R2$ and RMSEP (or RPD) of the PLS regression model obtained on the 35 independent*
*validation samples were judged acceptable, then the model may be used to predict the values of the*
*absolute content (g/kg) and proportion (%) of SOC in the POM and in the MOM fractions of the 2755*
*remaining topsoils of the German Agricultural Soil Inventory.*

**Answer: We agree that, if possible, the best method is always to have an independent validation**
**dataset. We think, however, that this is not advisable in our case, as the calibration dataset was for**
**the whole area of Germany, containing very different soils. In this case 145 samples are not a large**
**calibration dataset. This calibration dataset was selected out of all 2900 available soil samples**
**using the Kennard Stone algorithm, so that it contains the maximum possible spectral variability.**
**There were also additional selection criteria for these sites, as explained in ll.125-131. This is why**
**we do not want to split the reference dataset into calibration and validation dataset, as with every**
**split of this dataset a large part of the variation present in German soils would be lost for the**
**calibration.**

*We therefore argue that the PLS regression model based on near-infrared spectroscopy presented by*
*the authors cannot be used in its current form to predict labile and stable SOC fractions on "new"*
*topsoil samples of the German Agricultural Soil Inventory. At this stage (i.e. unreliable assessment of*
*the predictive performance of the PLS regression model), the authors can only use the reference data*
*(n = 145) of the absolute content (g/kg) and proportion (%) of SOC in the POM and in the MOM*
*fractions to investigate the potential drivers of the distribution of SOC kinetic pools on this limited*
*dataset. This would already be a significant piece of work.*

**Answer: As we conducted an independent validation, which showed that the predicted values are**
**in good accordance with the measured ones, we are sure that the model is robust enough and can**
**be used to predict the 2755 "new" samples. Therefore, we argue that the drivers can be assessed**
**not only using the reference data, but also the predicted ones.**

*Furthermore, Vos and colleagues used the particulate organic matter (POM) fraction to represent the*
*labile SOC kinetic pool. However, the POM fraction could contain substantial (and variable) amounts*
*of pyrogenic carbon with residence time in soils higher than the mean residence time of total SOC.*
*This limitation of the SOC density fractionation scheme should be mentioned and discussed in the*
*text, as it is not possible to guaranty that the POM fraction truly represents the actual labile SOC pool*
*for all investigated samples.*

**Answer: We agree that this is a limitation of density fractionation, which we will address in the**
**revised version of our paper. Pyrogenic carbon does, however, play a minor role in German soils.**
**There is also a large section on the so-called "black sands" in Germany (ll.300-356), where we**
**discuss explicitly why the POM fraction is not always a labile fraction.**

**B) Referee comment 1**

Dear anonymous referee,

We thank you for reviewing our manuscript and for giving instructive feedback on how to improve it.
We very much appreciate your work as reviewer. Please find our answers to your comments below:

L62: There is ample evidence that no-till does not lead to net increase of SOC com- pared to conventional tillage as indicated here, but only to a change of the depth distri- bution of SOC

**Answer: We agree with the reviewer that this should be mentioned more clearly and, thus, we will**
**include references to studies that report this depth distribution of SOC as a result of no-till (Baker**
**et al. 2007, AGEE, review from Luo et al. 2010, AGEE).**

Section 2.3: The fractionation approach is not really clear: to separate the fPOM, normally SPT is used as done in this study, but without any dispersion (as indicated by "free"). Here, ultrasonic dispersion at 65 J/mL was applied that probably de- stroyed macroaggregates, so the extracted POM is rather fPOM+oPOM (derived from macroaggregates). Of course you can do that, but this fraction should not be called fPOM. Furthermore, 450 J/mL was used to destroy "aggregates" (I guess microag- gregates), please explain why this energy level was used (reference). I further miss information on recovery rates of the fractionation and further basic data such as frac- tion mass and C content in order to evaluate the approach. Particularly the measured

C content of the POM is important to evaluate the fractionation approach.

**Answer: We see now that more details are needed in the manuscript concerning the fractionation**

**procedure. We used a very low dispersion energy of 65 J/mL to obtain the FPOM fraction. We did**

**this as in Don et al. 2009, JPNSS and other publications. Such a light ultrasonic treatments helps to**

**standardize the shaking of the samples that has been proposed in the original method by Golchin.**

**The energy level of 450 J/mL to obtain the OPOM fraction was chosen as Schmidt, Rumpel and**

**Kögel-Knabner (1999, European Journal of Soil Science, 50, 87-94) found that 450-500 J/mL is**

**enough to disperse all aggregates (including microaggregates) in a wide range of soil types. We will**

**include this reference to the revised version of the paper, as well as information on recovery rates,**

**mean fraction masses and C-contents of the fractions, which are indeed valuable criteria to**

**evaluate the fractionation approach.**

**We know that there is ample discussion on fractionation methods and how to obtain which**

**fractions, but we do not want to go into detail in this paper, as it is not the main focus and the**

**FPOM and OPOM fraction are merged for the NIRS prediction anyway.**

L182+L194: More information is needed in this regard, Jaconi et al. is not accessible (see also the comment of Lauric Cécillon). Please include relevant information also in this paper, even if Jaconi et al. is published during revision.

**Answer: As already stated in the reply to the comment of Lauric Cécillon we will include more**

**details on the NIRS calibration and validation approach into the supplement of the revised version.**

L188-198: I would rather see that as results

**Answer:**

**We propose not to put this paragraph in the results section, as it is the justification for using the**

**methodology and not the result and topic of this paper. But we changed this paragraph in the**

**revised version as follows:**

**"We used the methodology as stated above, as Jaconi et al (submitted) found out that NIRS is a**

**fast, low-cost and accurate method to predict the carbon fractions. The authors found the**

**following calibration results: For prediction of carbon content in the fractions (g kg-1), the**

**coefficient of determination ($R^2$) between predicted and measured carbon content in the fractions**

**was found to be 0.87-0.90 and RMSECV was 4.37 g kg-1. The RPD was 2.9 for the prediction of**

**carbon content in the light fraction and 3.2 for the prediction in the heavy fraction. For prediction**
**of carbon proportions in the fractions (%), $R^2$ was 0.83, RMSECV 11.45% and RPD 2.4 (Fig. S1 and**
**S2; for more details see Jaconi et al., submitted). The accuracy of prediction of both SOC content**
**and proportions of the light and heavy SOC fractions was very good and was comparable with that**
**in other studies that have used NIRS to predict SOC fractions (Cozzolino and Moro, 2006; Reeves et**
**al., 2006)."**

L197-198: NIRS is certainly a promising way to predict fractions, but of course this approach is specific to the fractionation. As there are numerous other fractionation ap- proaches (probably even better ones to derive "active" and "passive" SOC), this study should avoid giving the impression that the presented approach is the only way to esti- mate active and passive SOC at the regional scale.

**Answer: In l. 197-198 we merely aim to say that NIRS is a good way to predict the fractions, not**
**that it is the only way to do so. We will change the sentence accordingly.**

L203-205: More information is needed on the calculation of C and N inputs as well as on the regional yield estimates.

**Answer: We will include more information on the calculation of C and N inputs in a revised**
**manuscript.**

L229: In order to avoid interaction effects between the variables, one could perform

PCAs prior to the analysis and reduce the number of predictors to independent ones (e.g. dependent climate variables MAT, MAP and elevation can be reduced to one factor climate). For example, CaCO3 was identified as important, but this is probably only due to a correlation with texture (clay is the most important factor).

**Answer: The reviewer is right suggesting that using PCAs prior to the cforest analysis would reduce**
**the number of predictors to independent ones. We refrained from doing so however, as the cforest**
**algorithm did not find very many variables of a high importance in our case. With our approach we**
**receive a nonbiased assessment which is not influenced by a preselection of certain variables. We**

**therefore do not see the need to conduct the PCAs beforehand and decided to discuss all the single**
**variables, keeping in mind, of course, that a high variable importance can also be due to**
**interactions with other predictors.**

**We did however eliminate predictor variables with correlations above 0.8 from the dataset as to**
**avoid multicollinearity. We will therefore add the following sentence to the revised version of the**
**manuscript: "As multicollinearity between the predictors may result in a biased variable**
**importance measure in cforest algorithms, (Nicodemus et al., 2010) the correlations between the**
**predictor variables were controlled. When the correlation between two possible predictors was**
**>0.8, only the one with the broader range of variation was kept in the dataset."**

L316: remove "and"

**Answer: the "and" will be removed in the revised version. Thank you for noticing.**

Section 4.4: In principal, I agree that the fractionation approach based on a separation of POM from MOM is suitable to derive "labile" and "stable" carbon, as POM is the major constituent of "active" carbon (assuming that the contribution of pyrogenic carbon is negligible, which is the case in most regions of Germany). However, the authors could mention that there are other ways to derive labile and stable SOC.

**Answer:  We agree that there are very different methods/fractionation schemes to separate labile**
**from stable SOC. Therefore, we will add the following sentence: "The applied fractionation method**
**is only one out of several methods and options to separate labile from stabilised SOC."**

**C)  Referee comment 2/Short comment 2**

**Dear Dr. Smith,**

**Thank you for reviewing our manuscript so thoroughly and taking the time to write helpful and**
**detailed comments to improve our paper. We are very grateful for this.**

**Please find our answers to the comments below:**

Introduction

Overall, I think that the introduction needs some restructuring and needs more "meat"  to it. Many
statements are vague, blanket statement and don't provide much insight or examples (e.g. "The effects of land use and management are not the same for all soil organic matter compounds..." How?
Why? Give me more details). I think that the manuscript would benefit from a closer look at the flow
and organization of the introduction. I suggest taking a close look at each paragraph; map out the
main point, make sure this main point is reflected in the topic sentence, and verify that the preceding
and following paragraphs fit/flow. There are a few paragraphs that just don't fit (seem out of place)
and it detracts from the main points of the introduction (which is essentially to build up to, i.e.
provide background and rationale, the objectives and hypotheses of the study). As such, please align
the introduction specific to the goals and objectives of the study.

**Answer: We agree with the reviewer that in some cases more details need to be given in the**
**introduction. We also see now that a stricter alignment of the introduction with our research goals**
**would be helpful. We will follow this advice and restructure the introduction section in the revised**
**version of the manuscript.**

I strongly encourage the authors to reframe the objectives of the study as hypotheses in lieu of the
somewhat vague research questions that are currently reported in the introduction. What do the
authors expect the distribution of POM vs. MOM to be across Germany (and why)? Which factors
(land-use, climate, soil type, clay content, etc.) do the authors expect to be more important in driving
these distributional patterns? And the final question "can regions of high vulnerability..." needs to be
clarified. First, I don't know how you define "vulnerable" and second, I am unaware how you plan on
verifying that your predictive approach (i.e. machine learning)

**Answer: We agree that the third objective needs to be clarified and we will introduce the term**
**"vulnerable" before and be more explicit regarding the methodology. However, we refrain to**
**rephrase our objectives as hypotheses as the study design is not like in traditional studies that test**
**different treatments for which a hypothesis is formulated.**

Many of the statements or research addressed here are specific to European agroecosystems and yet
the authors often make broad statements about land use and management effects on SOC as fact.
However, land use and management effects on SOC differ greatly depending on cropping system,
location (climate, topography, parent material, etc.) and there is often an equal amount of work that
supports different results than what you present in this paper. As such, please be more specific and
make sure to constrain postulations with "in temperate cropping systems..." or something to that
example. I would be satisfied with a sentence early on stating that you are limiting the state of art (or
body of knowledge) to your specific system (i.e. western European cropping systems).

**Answer: The reviewer is right in that some statements in the introduction mainly refer to Western**
**Europe and we will follow her advice and state this early on in the introduction.**

As mentioned earlier, many sentences are vague. Please try to be more specific and detailed when
building up the background and rationale in the introduction. There is more "telling" than "showing".
Please see the attached line-by-line review.

**Answer: Thank you for uploading the commented version of the manuscript. We agree that the**
**revised version of the introduction must be more specific and detailed and will change it**
**accordingly.**

Methods

Overall, I suggest reorganizing the methods section to be more aligned with your objectives. This is
especially true when it comes to the use of calibration versus all samples. Sections often jump from
calibration to all and it makes it a bit confusing. There also needs to be more technical details into
how soils were collected and processed (e.g. collected with a corer, composite samples, one sample
per depth, homogenized, dried, etc. ?). Replication need to be explicitly stated (how many samples
did you use for each classification combination – i.e. land use, or depth, etc.). Including a
supplemental table that lists all the samples/sites or something may help clear this up. There are also
several areas where the methods need to be more explicitly stated and many instances were
citations are needed. Please see attachment for line by line comments.

**Answer: We can see that the methods section can be confusing for the reader in the present stage**
**and we will revise and improve it in the revised version. More details on the soil sampling and**
**handling will be included and methods will be described in more detail.**

Calibration samples versus all: The experimental design (use of calibration sites versus all sites) needs
to be clearer. It was confusing with the way the methods section was organized for the reader to
understand why/what/how calibration samples were used as compared to all sites. Perhaps have a
separate calibration section in the methods where all of this is addressed would be clearer.

**Answer: We agree that a separate calibration section is a good idea to clarify the methodology. We**
**will restructure the methods accordingly.**

A major issue I have with the methods is combining the oPOM and fPOM fractions together as a
"light fraction." As much as I hate to ask authors to redo their analyses, I think that the best way to
deal with the oPOM is to either ignore it or analyze it separately.

**Answer: We agree with the reviewer in the point that fPOM and oPOM are not the same. We have,**
**however, good reasons to combine them into a light fraction for the purpose of prediction:**

-    **The oPOM fraction generally constitutes only a very small part of total SOC (Mean: 4%).**
**Thus, it is very hard to predict this fraction separately via NIRS. We tried it as a first step**
**but calibration results were very poor. This is why we do not treat oPOM separately from**
**fPOM.**
-    **We do, however, not want to ignore the oPOM fraction completely for the following**
**reason: The novelty in the prediction of C-fractions via NIRS consists of using the log-ratio**
**transformation to ensure that the carbon content of both fractions adds up to 100% of the**
**total carbon content of the sample. Therefore, we cannot omit the oPOM fraction since it**
**would be unclear to which value the fPOM and MOM fraction should add up.**

Results

Please review my comments in the attachment and address them. Most importantly, I do not agree
with using total SOC to explain fraction SOC. Of course, C would explain C. Total SOC is NOT a driver –
it is a response variable for this study.

**Answer: We will address the helpful comments in the results section in the revised version of the**
**paper. We do, however, not agree that the SOC content is merely a response variable in our**
**dataset. The question needs to be answered whether the light and the stabilised fractions are**
**regionally so variable that they require a separate analysis and cannot be predicted from the total**

**SOC content. If total SOC content is a strong predictor for the fractions we could easily build a model to predict fractions from total SOC and do not need fractionation work. It is important to check whether and which of the fractions are closely related to total SOC, as this implies a higher relevance of this fraction for the total SOC content of the soil. For example, our results show that total SOC is much closer related to the light fraction in the black sands than in the other soils where texture is a more important driver for the distribution of the fractions.**

You are also missing any reference to Fig. 6 and Fig. 8 in the results! If you don't use them – don't put them in the manuscript (or put them in supplemental).

**Answer: Thank you for noticing this. We will include these references in the revised version of the manuscript.**

Discussion

I would almost reorganize the discussion to be more explicitly aligned with the study objectives – first discuss the how SOC is distributed among fractions at a national scale, then discuss which drivers are relevant and finally end with whether or not you can predict "vulnerable" (but please define) areas using your approach. Section 4.1 is entirely too brief, especially since it supposedly addresses your first objective. Again – don't just tell me what other results support or do not support your results, show me!

**Answer: We agree that section 4.1 should be more detailed and should show more results of other studies. We refrain, however, from restructuring the discussion as proposed by the reviewer for the following reason: In our first draft version, the discussion was structured exactly as proposed by the reviewer. There we encountered the problem, however, that there were alternating parts about black sands and "normal" soils which forced us to repeat the same information over and over. We therefore decided to structure the discussion into a "black sands" and a "normal soils" part.**

You have a great discussion on the "black sands" section. I would love to see that reflected in the entire discussion section. Some of the details I was looking for in section 4.1 are included in 4.2. I think it would be good to combine section 4.1 and 4.2 (and address your first objective) and discuss black sands in the context of objective 1.

**Answer: We agree that it would indeed be a good idea to combine these sections in the revised version.**

In section 4.4, it would be great to discuss why/why not you think your approach worked to identify vulnerable areas. It is one of your objectives and you do not directly discuss it in the discussion. It needs to be addressed. I think concluding section 4.4 with a paragraph answering "Can regions of high vulnerability to carbon losses be identified by this predictive approach?" is warranted.

**Answer: We also agree with this proposal and will enhance the discussion of our third objective accordingly.**

Conclusion

See notes regarding final sentence. I believe that with a few revisions (as per my and other reviewers'
suggestions) this manuscript is publishable and I look forward to the revisions!

**Answer: We will reformulate the last sentence to make it more specific in the revised version.**

**D) Short comment 3**

Dear Dr. Viscarra Rossel,

Thank you for your short comment regarding our manuscript. We very much appreciate your input
that helps to improve our paper and to make it more clear and easy to read. Please find our answers
to your suggestions below:

I thank the authors for their paper and I hope that my discussion helps. My comments here relate
primarily to the lack of clarity in the description of the methods used for the spectroscopic modelling,
and to missing quantification of robustness and uncertainty in the spectroscopic model predictions of
the carbon fractions. I believe these to be crucially important because their further analyses and
interpretation of the variability and driving factors relies heavily on the spectroscopic model
predictions.

First, the description of the spectroscopic modelling is inadequate and I encourage the authors to
improve it. I think that the specifics of the spectroscopic modelling, apparently described in Jaconi et
al., need to be included in this manuscript, particularly because the Jaconi et al. manuscript isn't yet
published. But, even if the Jaconi et al paper were published, I think that at the very least, readers
will need a clear summary of their methods and findings–not simply a report of their assessment
statistics.

**Answer: We agree that the reader needs more information on the spectroscopic modelling and as**
**we are not sure when the review process for the paper of Jaconi et al. will be finished, we will**
**include a more detailed description in the methods section of the revised version.**

Second, the authors do not convincingly show that the spectroscopic models were sufficiently robust
for predicting the 'unknowns', which I presume were the '...>2500 sites with mineral soil all over
Germany' (mentioned only in the Introduction, line 106). Additional validation of the models with an
independent test set will help, however, I would also encourage the authors to implement either a
repeated cross validation, or to bootstrap the models to quantify their robustness and the
uncertainty of their predictions (see for instance Viscarra Rossel, 2007). To this end, the authors
might find it useful to read Viscarra Rossel & Hicks (2015). There, we proposed an approach for
modelling the carbon fractions of a large continental scale dataset, reporting the robustness of the
models, the (propagated) uncertainties of the predictions, and relating the spectroscopy to the
chemistry of soil organic C.

**Answer: As described in our answer to the comment of L. Cécillon, the models have been validated**
**using an independent test set and the results will be included in the revised version of the**
**manuscript. Both datasets, the calibration and the validation data set cover the area of interest**
**(Germany). We will check the recommended papers for the options to further quantify the model**

**uncertainty. However, with an independent validation dataset we already quantified the model**
**uncertainty.**

Quantifying uncertainty is particularly important when predicting 'unknown' samples. Without
quantified uncertainty, the predictions will definitely be less valuable. This is particularly relevant for
this study because the predictions are being used in subsequent analysis to potentially gain new
understanding.

**Answer: We agree that the quantification of uncertainty is crucial for gaining trust in the predicted**
**values. Therefore, we propose to include a summary of the calibration and validation results in the**
**supplement material of the revised version.**

Finally, I would like to suggest some minor corrections:

- In lines 182–183, the Jaconi et al reference is cited as 'in prep' while in line 194 it is cited as
'submitted'

**Answer: Thank you for noticing this mistake. We will change this in the revised version of the**
**manuscript. However, we hope to get this paper to be published soon.**

-The mention of the'...>2500 sites with mineral soil all over Germany.', in the Introduction, line 106,
is inadequate. This should be described and made clear in the Methods section–possibly in section
2.4 after a (better) description of the spectroscopic modelling.

**Answer: We agree that the methods need to be clear. However, there is a section on the soil**
**inventory (2.1) and we will add more in the spectroscopic method section. In this case we do not**
**agree with the comment, as it is good practice to give a very short overview in the introduction on**
**how the research questions shall be answered. Of course the number of sites should also be stated**
**in the methods section, which is the case.**

- In lines 185–187: '... In addition, residual prediction deviation (RPD) was calculated, using the
classification system devised by Viscarra Rossel et al. (2006)....'

– I am quite sure that Viscarra Rossel et al. (2006) did not devise a classification for the RPD. Williams
(1987) originally devised the RPD for assessing spectroscopic calibrations of agricultural and food
products. Later, Chang et al. (2001) suggested an arbitrary classification specifically for soil. It is very
likely that Viscarra Rossel et al. (2006) simply used that classification, but I could not confirm one way
or the other because the Viscarra Rossel et al. (2006) reference is not listed in the references.

**Answer: Thank you for this clarification. We will revise this and change it to Chang et al.**
**mentioning that the classification is arbitrary but can serve as indicator for the model quality.**

- In terms of the RPD, Bellon-Maurel et al. (2010) suggested that the RPD should only be used if the
data is normally distributed, otherwise, they propose the use of the RPIQ (Bellon-Maurel et al.,
2010).

**Answer: We will also include the RPIQ in the revised version.**

- Following from that, in our spectroscopic modelling of soil carbon and fractions (Viscarra Rossel &
Hicks, 2015), we found that their statistical distributions were often not normal and required logarithmic transformations. For this reason, it would be useful for the authors to report the distributions of the carbon and fractions data–but also because the PLSR algorithm assumes normally distributed data.

**Answer: We agree with this and we log-transformed the data for model development. We will add information on this in a revised manuscript version.**

**E) Editors comment**

When submitting a revised manuscript please ensure that you address the following points:

- SC1 and RC1 raise an important point about the Jaconi et al not being available at this point. The authors should update on the status of the paper.

**Answer: In the revised version we attached additional information on the NIRS calibration and validation in the methods section, and the calibration and validation results obtained for the present dataset by Jaconi et al. in an additional supplement. The status of the paper of Jaconi et al. was updated to "in review".**

- Response to RC1: Page 2, the reviewer raises important point about not calling a pool of OM extracted after ultrasonication as fPOM. I appreciate the author's response and additional info that they are providing in the revised manuscript. But, it is still important not to refer to the OM extracted using ultrasonication as fPOM. Please revise the text.

**Answer: We see now that for some readers the term fPOM for the obtained fraction can be confusing. We changed this to the term iPOM (intra-aggregate POM). The text in the revised version was changed accordingly.**

- I agree with RC3 that putting both fPOM and oPOM pools together is problematic. These two pools (even though they can be very small fraction of soil C) differ in their availability for decomposition, and hence persistence in soil. Even if it is difficult to predict oPOM alone, and if the authors have a hard time achieving good results when they treat the two pools (as stated in C4) it is important to make sure that adding these two pools is not leading to confounding and potentially misleading results.

**Answer: We agree that the fPOM and oPOM pools differ in their availability for decomposition, but we still think that combining both fractions for the purpose of prediction at a national scale is the way to go in our special case: As we wanted to obtain the best prediction, treating fPOM and oPOM separately was not an option, as oPOM was not reliably predictable due to its small proportions in German agricultural soils. Leaving out the oPOM fraction was also not possible as all fractions should up to 100% when using the log ratio.**
**We do not think that the results obtained in this way are confounding or potentially misleading, as it is clearly stated that the light fraction contains both fPOM and oPOM. On top of this, one main focus of the whole paper is that the light fraction is not necessarily a labile fraction, due to the occurrence of black sands in Germany. This finding makes it clear again that the fractions are only defined operationally and do not always imply a good measure of the carbon residence times in the soil. Soil organic matter pools and fractions are arbitrary defined (or operationally defined) except for the difference between POM and SOM that is bound to the mineral phase. Difference in stability between these two SOC pools has been confirmed in hundreds of studies. Our**

**fractionation scheme aimed at separating these two pools and additionally separated POM in two**
**fractions. However, the main difference is between the POM fractions and the MOM.**

**2) Author's changes in manuscript**

[revised manuscript text omitted]
 input aboveground and belowground input of SOC (Christensen, 2001). This The smallerlimited differences between in light fraction between in cropland and grassland soils shown in our study may partlycan possibly be due to interfering factors, asdue to historical land useland-use changes which would need deeper investigations to unravel.conversion of cropland to grassland still affecting carbon distribution in the fractions. Moreover, Ggrasslands and croplands are often generally located on different soil types which, again, interferes with other factors as soil moisture or texture. , however; and thus tTherefore, it is not always possible to draw direct conclusions on land-use change effects on carbon fractions from such regional inventories. In a previous study using paired land-use change sites, the POM proportion was found to be twice as high in grasslands as in croplands (Poeplau and Don, 2013b). Even though the fraction distribution did not differ significantly between croplands and

[revised manuscript text omitted]

---

## Editor Decision (ED1)

**Hot regions of labile and stable soil organic carbon in Germany - Spatial**

**variability and driving factors**

Cora Vos[1], Angélica Jaconi[1], Anna Jacobs[1], Axel Don[1]

[1]Thünen Institute of Climate-Smart Agriculture, Bundesallee 50, 38116 Braunschweig, Germany

Corresponding Author: Axel Don, axel.don@thuenen.de, Tel. +49 531 596 2641

Keywords: Soil organic carbon fractions, near-infrared spectroscopy, NIRS, soil carbon stability,
National Soil Inventory, German Agricultural Soil Inventory, carbon sequestration

Notes, from AE Berhe
- throughout the manuscript, please be careful about usage of terms such as C content (i.e. absolute amount, in units such as g or Kg) vs. concentration (in units of % or g/kg). I notice multiple places, including axes labels on figures where g/kg values are reported as C content. Note that, after a while, I stopped highlighting where C content looks like it is being used where you should use concentration.

- the Jaconi et al in review period has to be taken out of here if it is not published already. Please restate all the justifications from that paper you would like to use here, in this manuscript as it is not ok to cite a paper that is not published.

[revised manuscript text omitted]

As pointed out by reviewers, if this paper is not accepted already, you have to take it out and restate the findsings here

We used the methodology as above described as Jaconi et al (in review) found that NIRS is one promising method to predict carbon fractions, which is fast, low-cost and accurate. The authors had the following calibration results: For prediction of carbon content in the fractions (g kg$^{-1}$), the coefficient of determination (R²) between predicted and measured carbon content in the fractions was found to be 0.87-0.90 and RMSECV was 4.37 g kg$^{-1}$. The RPD was 2.9 for the prediction of carbon content in the light fraction and 3.2 for the prediction in the heavy fraction. For prediction of carbon proportions in the fractions (%), R² was 0.83, RMSECV 11.45% and RPD 2.4 (Fig. S1; for more details see Jaconi *et al*., in review). The accuracy of prediction of both SOC content and proportions of the light and heavy SOC fractions was very good and was comparable with that in other studies that have used NIRS to predict SOC fractions (Cozzolino and Moro, 2006; Reeves et al., 2006).

**2.5 Drivers of soil organic carbon distribution in fractions**

A total of 75 potential drivers of differences in carbon proportions in different fractions was compiled from the soil analysis data, complemented with data from a farm survey and geographical data (for a complete list of predictors, see Table S2). The farm survey recorded management practices, over the

10 years, if known by the farmer, prior to sampling. Using this, yearly mean carbon and nitrogen inputs through plant material and organic and mineral fertilizers were calculated for each site based on the yield of the main product and on different carbon allocation functions for different crops as described in (Bolinder et al., 1997)When data were missing in the survey responses, yields were calculated using regional yield estimates provided by the regional governments. Climate and site data acquired from GIS data layers completed the set of predictor variables (climate data from Deutscher

Wetterdienst, normalised difference vegetation index (NDVI) data from ESA, elevation data from

Bundesamt für Kartographie und Geodäsie). For the sites in the federal states of Lower-Saxony,

North-Rhine Westphalia, Mecklenburg-Western Pomerania, Rhineland-Palatinate, Saxony Anhalt and

Schleswig Holstein (Northern Germany), the land-use history was researched using historical maps (dating back to 1873-1909), as many regions in these states are known to have a heathland or peatland legacy.

The conditional inference forest algorithm (cforest; Hothorn *et al.*, 2006), was used to identify the most influential drivers of SOC distribution among the different fractions. Cforest is an ensemble model and uses tree models as base learners that can handle many predictor variables of different types and can also deal with missing values in the dataset (Elith et al., 2008). The cforest algorithm is similar to the better known random forest algorithm, a non-parametric data mining algorithm that uses recursive partitioning of the dataset to find the relationships between predictor and response variables (Breiman, 2001).

Bootstrap sampling without replacement was carried out in order to prevent biased variable importance (Strobl et al., 2007). As multicollinearity between the predictors may result in a biased variable importance measure in cforest algorithms (Nicodemus et al., 2010), the correlations between the predictor variables were controlled. When the correlation between two possible predictors was > 0.8, only the one with the broader range of variation was kept in the dataset. Ten cforest models were created, each containing 1000 trees and using different random subset generators. From these models, the variable importance of predictors was extracted and the relative variable importance was calculated and averaged over all 10 models. Variables were considered important when their relative variable importance was higher than 100/n, where n is the number of predictors in the model. This is the variable importance that each variable would have in a model where all variables are equally important (Hobley et al., 2015). It should be noted that the relative variable importance value obtained from the cforest algorithm does not necessarily imply direct relationships between the proportion of SOC in the light fraction and the main drivers, as the algorithm also takes into account interaction effects between the variables. Model performance was assessed by the coefficient of determination ($R^2$), as defined by the explained variance of out-of-bag estimates, which represent a validation dataset:

$$R^2 = 1 - \frac{MSE_{OOB}}{Var_z} \qquad (1)$$

where $MSE_{OOB}$ is the mean squared error of out-of-bag estimates and $Var_z$ is the total variance in the response variable.

A range of soils in northern Germany, called 'black sands', behaved quite differently from other soils in the country in terms of the driving factors for SOC distribution among the fractions. Therefore the dataset was split into two parts for the cforest analysis and the cforest algorithm was used on: 1) the dataset containing only the black sands from northern Germany (n=264). Those were extracted using the NIR spectra, which were classified into black sands and normal soils using the simca function in the "mdatools" package (Kucheryavskiy, 2017); and 2) on all other soils considered not to be black sands (n=2406). All statistical analyses were conducted using the software R . Maps were generated with the software QGIS.

**3 Results**

**3.1 Carbon distribution among measured fractions (145 calibration sites)**

TheiPOM fraction contributed an average of 23% to bulk SOC (23% ±2.36 (mean ± standard error (SE)) in croplands and 25% ±3.8 in grasslands (Fig. 1). The oPOM fraction accounted for an average of 4% of SOC (3% ± 0.5 in croplands, 8% ±1.3 in grasslands) across all calibration sites (Fig. 1). The heavy fraction contributed the highest proportion to bulk SOC (73% in all soils, 73% ± 2.5 in croplands and 68% ± 4.4 in grasslands). The differences between land-uses were not significant. There was great variation in the carbon distribution between fractions, with the iPOM fraction contributing between 3 and 99% to bulk SOC. The absolute carbon content (g kg$^{-1}$) of the fractions was significantly different for the heavy fraction, with grasslands having significantly higher heavy fraction carbon content than croplands (31 g kg$^{-1}$ ± 3 compared with 13 g kg$^{-1}$ ± 0.7).

*[margin note: Please clarify this statement, what does landuse not affect, %C or distribution of C in fractions or both?]*

[revised manuscript text omitted]

---

## Author Response (AR2)

Dear Asmeret,

Dear Editors,

Thank you very much for the positive decision on our paper.

We corrected the last bits and pieces according to your recommendation.

We agree with you that the use of terms need to be consistent throughout the manuscript. However, we do not agree with the use of the term concentration for SOC in g/kg. According to the international definitions of units "concentration" refers to mass per volume. Therefore, we propose to keep the term "content" for a mass fraction of SOC in soils since this value does not refer to a fixed volume.

We hope you can accept this since we checked that the terms are used consistently throughout the manuscript.

Best regards,

Cora Vos and Axel Don

[revised manuscript text omitted]